# Numerical and Fracture Mechanical Evaluation of Safety Monitoring Indexes and Crack Resistance in High RCC Gravity Dams Under Hydraulic Fracture Risk

**DOI:** 10.3390/ma18122893

**Published:** 2025-06-18

**Authors:** Mohamed Ramadan, Jinsheng Jia, Lei Zhao, Xu Li, Yangfeng Wu

**Affiliations:** 1Key Laboratory of Urban Underground Engineering of Ministry of Education, School of Civil Engineering, Beijing Jiaotong University, Beijing 100044, China; 20119903@bjtu.edu.cn (M.R.); xuli@bjtu.edu.cn (X.L.); 2State Key Laboratory of Simulation and Regulation of Water Cycle in River Basin, China Institute of Water Resources and Hydropower Research (IWHR), Beijing 100038, China; 3Huadian Electric Power Research Institute Co., Ltd., XiYuan, Road 1-10, Xihu District, Hangzhou 310030, China; 526947211@163.com; 4State Key Laboratory of Hydraulic Engineering Simulation and Safety, Tianjin University, Tianjin 300072, China; wuyangfeng_0211@tju.edu.cn

**Keywords:** high concrete RCC gravity dams, hydraulic fracture, safety monitoring index, FEM, fracture mechanical model, FAD, crack propagation

## Abstract

High concrete gravity dams, particularly Roller-Compacted Concrete (RCC) types, face long-term safety challenges due to weak interlayer formation and crack propagation. This study presented a comprehensive evaluation of safety monitoring indexes for the Guxian high RCC dam (currently under construction) using both numerical and mathematical models. A finite element method (FEM) is employed with a strength reduction approach to assess dam stability considering weak layers. In parallel, a fracture mechanical model is used to investigate the safety of the Guxian dam based on failure assessment diagrams (FADs) for calculating the safety factor and the residual strength curve for calculating critical crack depth for two different crack locations, single-edge and center-through crack, to investigate the high possible risk associated with crack location on the dam safety. Additionally, the Guxian dam’s resistance to hydraulic fracture is assessed under two fracture mechanic failure modes, Mode I (open type) and Mode II (in-plane shear), by computing the ultimate overload coefficient using a proposed novel derived formula. The results show that weak layers reduce the dam’s safety index by approximately 20%, especially in lower sections with extensive interfaces. Single-edge cracks pose greater risk, decreasing the safety factor by 10% and reducing critical crack depth by 40% compared to center cracks. Mode II demonstrates higher resistance to hydraulic fracture due to greater shear strength and fracture energy, whereas Mode I represents the most critical failure scenario. The findings highlight the urgent need to incorporate weak layer behavior and hydraulic fracture mechanisms into dam safety monitoring, and to design regulations for high RCC gravity dams.

## 1. Introduction

Concrete gravity dams are critical infrastructure for water resource management, flood control, and hydropower generation. Ensuring their long-term safety and stability is essential, as failure can result in catastrophic consequences. However, traditional monitoring methods often fail to capture the complex mechanisms contributing to dam instability, particularly under long-term loading conditions. These challenges are more pronounced in Roller-Compacted Concrete (RCC) dams, which are prone to the formation of weak layers due to their layer-by-layer construction process. Over time, these weak interfaces can compromise the dam’s structural integrity and increase the risk of progressive failure. In high concrete gravity dams exceeding 200 m in height, another critical risk factor is hydraulic fracture, which can negatively impact dam safety by inducing crack propagation under high upstream water pressure [1]. To contextualize these risks, the Guxian high RCC dam, currently under construction in China, serves as a valuable case study for this research, as the Guxian dam will stand at a height of 215 m. Given its size and construction method, the dam faces potential risks from both weak layers and hydraulic fracturing. Investigating these two key failure mechanisms is crucial for ensuring the long-term safety and stability of high concrete gravity dams. Historical incidents further highlight the relevance of this issue. Several cases of concrete dams damaged by hydraulic fracture have been documented. The Kölnbrein arch dam in Austria, standing at 200 m, experienced a sudden leak during its third storage period. This was attributed to crack expansion at the dam heel caused by hydraulic fracture and increased uplift pressure [2,3]. Similarly, the Zilancolente arch dam, with a height of 182 m, had a bottom joint installed at the foundation of the upstream surface to mitigate hydraulic fracturing. However, after impoundment, cracks formed above the bottom joint due to hydraulic fracturing [4,5]. In China, the Zhexi concrete buttress dam, 104 m tall, suffered upstream face cracks due to high water pressure. Other incidents include cracks in flood discharge tunnels [6,7,8], such as at the Dongfeng Arch Dam [9], and penetrating cracks caused by poor temperature control after impoundment, as seen in the massive head dam of Tuoxi [10]. In the United States, the 219 m Dworshak gravity dam encountered cracks and leakage due to hydraulic fracture. These cracks appeared during its seventh year of operation, when the dam reached its maximum design water level, causing further crack propagation and spreading [11].

In light of these failures, catastrophic dam failures have highlighted the need for robust monitoring and maintenance strategies. Conventional monitoring methods, including periodic visual inspections and sensor-based data collection, provide valuable insights but may not adequately address the complexities of dam behavior under long-term loading and environmental stress. Conventional methods, such as visual inspections and sensors (e.g., piezometers, strain gauges), offer useful data but often fail to detect early-stage subsurface damage, crack propagation, or complex time-dependent behaviors. These limitations reduce their effectiveness in capturing long-term degradation and hydraulic fracture risks in dams. Therefore, continuous monitoring of both the dams and their foundations is essential for detecting abnormalities and analyzing the potential failure modes. The safety monitoring index (SMI) serves as a key tool to ensure that dams remain within acceptable operational limits, with the SMI representing the maximum value of monitored parameters before failure occurs. Various studies have focused on improving dam safety monitoring. Building on these efforts, Wang et al. [12] emphasized the need for reliable dam safety monitoring systems that assess the reliability of monitoring facilities, operation, and maintenance effectiveness. Suwatthikul et al. [13] introduced the Dam Safety Remote Monitoring System (DS-RMS) to enhance monitoring, aligning with international recommendations. Garcia et al. [14] assessed dam safety using weighted criteria to prioritize risk analysis. Gu et al. [15] created advanced monitoring techniques for critical areas of concrete arch dams to improve early-warning effectiveness. Jun et al. [16] introduced ensemble learning to improve dam safety monitoring, reducing the risk of misjudging abnormal data points. Wang et al. [17] proposed new safety monitoring indexes for high concrete gravity dams based on potential failure modes identified through finite element analysis. Their study addresses the limitations of traditional methods by assessing progressive instability using indicators like dam displacement and concrete yield zones. Monitoring indexes for dam stress, uplift pressure, and seepage are set according to hydraulic specifications and are typically established for critical local areas requiring strict control. Particularly for dam stress, the stress-controllable areas, like the dam heel and the toe, are also stress concentration areas identified by the finite element method [18]. However, as for crack opening, it is difficult to determine a monitoring index using hydraulic specifications due to the complex mechanisms involved. These mechanisms include tensile stress concentration, microcrack initiation and coalescence, aggregate interlock degradation, and the evolution of the fracture process zone (FPZ), which are highly nonlinear and sensitive to material heterogeneity [19,20]. However, like other monitoring items, local damage may occur when the monitoring index for crack opening is exceeded, and significant abnormalities will be reflected in dam deformation. Deformation is often the most indicative of overall dam safety and progressive failure [19]. Given this context, crack formation in concrete dams is a major concern, particularly as cracks compromise structural integrity and affect dam behavior. Cracks can develop during both construction and operation, posing a significant risk to long-term dam stability [21,22]. Additionally, it is important to develop methods for the timely detection of abnormalities and early warning signs, enabling the effective and reliable implementation of scientific and reasonable emergency plans to prevent major accidents [23,24,25]. To improve index accuracy, some researchers have used mathematical statistics or structural numerical calculation methods to estimate the monitoring index [26], while others have combined both methods to determine the monitoring index [27,28]. However, there are few reports on determining the monitoring index for concrete cracks. Although previous studies have addressed crack cause and stability analyses, and safety monitoring of hydraulic concrete structures, they often focus on a single perspective as a reason for crack propagation. Few studies integrate crack cause and stability analysis, making it difficult for the operation and maintenance personnel to take effective measures. A more integrated approach was explored by Huang et al. [29], who investigated the structural safety of a gate storehouse at the Wang Fuzhou Water Conservancy Project. The safety was evaluated using an integrated approach involving crack cause analysis, stability assessment, and online monitoring. A multi-point anomaly recognition model was developed by [30] combining an improved local outlier factor (LOF) with mutual validation, considering spatio-temporal correlations in dam monitoring data. Applied to real displacement data from concrete and rockfill dams, the model showed high accuracy and robustness, enhancing dam safety assessment. Future work will focus on handling more complex data and post-outlier processing. Advances in technology have also contributed to this field. Another study by [31] presented the development of an intelligent monitoring system for the Three Gorges Dam, aimed at overcoming key technical challenges through the integration of AI, IoT, big data, and GIS + BIM technologies. The resulting platform enables real-time data aggregation, digital twin modeling, online structural analysis, intelligent hazard prediction, and digital inspection workflows. These innovations significantly improve efficiency in safety management and establish a model for the digital transformation of large-scale hydraulic infrastructure. In addition, Huang et al. [32] proposed a precision single-point BeiDou positioning system to enhance the accuracy of dam safety monitoring. Using data from a dam in Sichuan, comparative tests showed that this system achieved significantly higher accuracy (up to 94.83%) compared to traditional satellite positioning. The results demonstrate that the BeiDou-based system offers reliable, long-term monitoring for dam displacement and overall safety. Further refinement of monitoring techniques was demonstrated by Zhang et al. [33], who introduced a new method for determining safety monitoring indices (SMIs) in RCC dams by incorporating seepage–stress coupling and material anisotropy. Using a finite element model in COMSOL, the results showed that this approach reduced seepage and displacement SMIs by 34.78% and 31.98% compared to traditional methods. The findings highlight the importance of accounting for coupling effects and anisotropy in accurate dam safety assessments. Motivated by these developments, the current study employed both numerical and theoretical models to develop comprehensive safety monitoring indexes that account for material properties and operational conditions. The numerical model, based on the finite element method (FEM) combined with the strength reduction method, is used to calculate the dam’s safety factor with and without the presence of weak layers. Meanwhile, the theoretical model utilizes a fracture mechanical evaluation approach, incorporating the failure assessment diagram (FAD) and residual strength curves to determine the safety factor and critical crack depth. Together, these models provide a robust framework for assessing dam stability by quantifying key indicators of potential failure.

Despite the progress made, the influence of weak layers on the safety of RCC dams has received limited attention in the literature. Only a few studies have explored their impact, yet the findings consistently indicate that weak layers pose a serious threat to dam stability and overall safety. For example, ref. [22] analyzed the combined effects of weak layers and crack propagation, revealing a significant reduction in the dam’s safety factor due to the presence of weak layers. Similarly, ref. [34] investigated the seismic response of RCC gravity dams by modeling weak construction layers using zero-thickness cohesive elements and a concrete damaged plasticity model. Validated with data from the Koyna Dam, the method was applied to the Guandi Dam to examine how weak and well-bonded layers at different elevations influence seismic behavior. The results showed that weak layers substantially reduce seismic resistance and alter failure modes, while even well-bonded layers contribute to increased displacement, energy dissipation, and damage under strong earthquakes. As a response to these concerns, this study introduces a dam safety monitoring index to better capture and highlight their influence on the overall stability behavior of RCC dams.

Following this analysis, another investigation was presented to study the Guxian dam resistance to hydraulic fracture under different mechanical failure modes. Few studies have discussed the behavior of different modes of failure in the fracture mechanical evaluation model, such as ref. [35], that considered the assumption of equivalent fracture energies for both modes, while refs. [36,37] have stated that the fracture energy in Mode II is typically 20 to 25 times greater than in Mode I. This includes the energy needed to form the inclined tensile microcracks within the fracture process zone (FPZ), as well as the energy required to overcome the shear resistance resulting from aggregate interlock and other asperities on the rough crack surfaces behind the crack tip. To address this knowledge gap, a comprehensive comparison between these two modes was carried out to investigate the resistance of the concrete dam to crack propagation during both modes. Fracture mechanical models are indispensable for analyzing the structural integrity and failure mechanisms of high concrete dams. These models predict crack initiation and propagation due to factors like thermal stress, static loads, and dynamic events such as earthquakes. Dams, including gravity, arch, and RCC dams, can fail in several modes, including Mode I (opening), Mode II (shear), and Mode III (tearing). Mode I, involving tensile stress, is the most common, while Mode II occurs due to shear stress, often during seismic events or foundation movement. Mode III involves tearing, typically caused by torsional or dynamic loading, leading to potential misalignment and block rotation. Understanding these failure modes is critical for assessing dam safety.

In summary, despite extensive research on dam safety monitoring and fracture mechanics, limited investigations have considered the combined influence of construction-induced weak layers and hydraulic fracture risks on long-term dam stability. This study fills these gaps by integrating a comprehensive numerical model and a theoretical approach to assess safety monitoring indexes for high RCC gravity dams. The proposed framework uniquely incorporates the effects of weak layers and provides a comparative assessment of crack propagation behavior and Guxian dam resistance to hydraulic fracture under both Mode I and Mode II. This dual-mode evaluation offers a more realistic and robust basis for dam safety assessment, especially for modern high RCC dams like the Guxian dam. This work contributes to advancing the state of the art by providing new insights into failure mechanisms and supporting the development of more accurate safety monitoring criteria for design and operational practices.

Based on these motivations and the gaps in the literature, the main goals and objectives of this research are as follows: to establish a safety monitoring index for the Guxian high RCC dam, considering the impact of weak layers and crack propagation on dam safety during long-term operation; to compare the behavior of two different crack locations during hydraulic fracture; and to evaluate the safety index during crack propagation. Additionally, this study examined the Guxian dam’s resistance to hydraulic fracture under different fracture mechanics failure modes (Mode I and Mode II) by evaluating the ultimate overload coefficient, both with and without considering uplift pressure.

## 2. Methodology

The sequence of this research is as outlined as follows: (1) evaluating the safety monitoring index for high RCC gravity dam with the effect of the weak layers using strength reduction method utilizing numerical modeling (FEM); (2) based on the fracture evaluation mechanical model, the safety monitoring index was evaluated for the dam comparing two different crack locations using the safety factor of failure assessment diagram (FAD) and the critical crack depth from residual strength curve; (3) a comparison between two different modes of fracture mechanics failure Mode I (open type) and II (in plan shear) of fracture mechanical model to investigate the dam safety against hydraulic fracture.

### 2.1. Material Strength

Material degradation and external environmental loads are the primary failure risks for concrete gravity dams. Among the external loads, the most critical are hydrostatic pressure from reservoir water, uplift pressure, and seepage forces, followed by thermal loads (e.g., temperature gradients), seismic loads, and ice pressure in colder regions. To address these risks, the strength reduction method was used to analyze failure mechanisms and progressive instability using ABAQUS 6.14 software. This method establishes safety monitoring indexes based on failure risks and specific dam conditions. The proposed framework combines FEM analysis with the strength reduction method to simulate loading scenarios, to evaluate displacement patterns, and to identify potential structural instabilities. Incorporating weak layers in RCC dams enhances the understanding of dam behavior and failure mechanisms.

#### Safety Monitoring Index Calculation Using Strength Reduction Method

The strength reduction method (SRM) is a widely used technique to assess the stability of concrete dams with complex geometries and material behaviors [17,18]. It progressively reduces material strength parameters, such as cohesion (*c*) and internal friction angle (*φ*), until failure occurs, thereby determining the safety factor. The reduction factor, known as the strength reduction factor (SRF) or the safety factor (SF), is calculated based on displacement limits or the formation of a continuous plastic zone, indicating structural failure. The strength reduction factor (SRF) is defined as follows:(1)$$SRF=\left(\frac{\tan\varphi^{\prime }}{\tan{\varphi^{\prime }}_{f}}\right)=\left(\frac{c^{\prime }}{{c^{\prime }}_{f}}\right)$$
where *φ’_f_* and *c’_f_* are the effective stress strength parameters at failure (reduced strength). The strength reduction method accounts for weak layers in RCC dams, arising from construction processes. The ultimate strength reduction ratio, defined as the safety factor, is reached when a penetrating plastic zone forms, destabilizing the structure through the gradual reduction of shear strength in the soft interlayers [17,18]. To model material behavior under stress, this method is combined with the Drucker–Prager model, commonly used for concrete and rocks due to its smooth surface failure and correlation with hydrostatic stress [18,38]. An isotropic, elasto-plastic model with linear softening characteristics was used to analyze the dam’s material behavior, with the Drucker–Prager criterion serving as the primary yield function [17]:(2)$$F=\alpha I_{1}+\sqrt{J_{2}}-K$$
where *F* is the yield function, and $\;I_{1}\;$ and $J_{2}$ are the first invariant of stress tensor and the second invariant of deviatoric stress tensor, respectively. Both *α* and *K* are positive constants that depend on the shear strength parameters of the material (cohesion “*c*” and friction angle “*φ*”). Both constraints *α* and *K* can be calculated as follows [17]:(3)$$\alpha =\frac{\;\sin\varphi }{\sqrt{3}\;\sqrt{(3+{\mathrm{Sin}}^{2}\varphi )}}$$(4)$$K=\frac{3\;C\;\cos\varphi }{\sqrt{3}\;\sqrt{(3+{\mathrm{Sin}}^{2}\varphi )}}$$

### 2.2. Fracture Mechanics

To simulate the Guxian dam, different plates dimensions with the same material properties of the Guxian dam were utilized to assign the safety monitoring index. Fracture mechanics describes the behavior of solids or structures with geometric discontinuities at the structural scale, combining crack mechanics and material properties [39]. Various techniques analytically solve the stress, strain, and displacement fields near a crack in an elastic solid. The stress function is shown in Figure 1. The stress intensity factor (*K*) quantifies the stress at the crack tip, calculated using limit value formulas derived from stresses or displacements near the crack tip, often through extrapolation [39]. *K* predicts the stress state around the crack tip due to remote loads or residual stresses. It depends on the applied stress, crack size, and geometry, and varies with the crack opening mode, as demonstrated by Equation (5):(5)$$K=Y\;\sigma \;\sqrt{\pi a}$$

The geometry factor (*Y*) represents the crack system’s geometry relative to the applied load, and affects the stress intensity factor at the crack tip.

For a center crack in an infinite plate, *Y* = 1.0. For an edge crack in a finite-width strip, the correction factor depends on (*a/w*). This study considers a single edge-through crack in a semi-infinite body and a center-through crack in an infinite body. Static fracture analysis assumes a constant applied load and uses linear elastic fracture mechanics (LEFM) to compare the stress intensity factor (*K*) with the material’s critical value, typically the plane–strain fracture toughness (*K_IC_*). The stress intensity factor is calculated as shown in Table 1.

Linear elastic fracture mechanics (LEFM) assumes the elastic material behavior, with a small plastic zone near the crack tip compared to the overall part. If the plastic zone grows large and reaches boundaries, the linear elastic assumption becomes invalid, indicating gross yielding. A plastic zone forms ahead of the crack tip, and the elastic stress field equations are used to determine the theoretical distance from the crack tip where the stress equals the material’s yield strength. This involves solving the elastic stress field equation to find the point where the stresses match the yield strength. The elastic stress field equation used here is as follows:(6)$$\sigma_{y}=\frac{K_{app}}{\sqrt{2\pi r}}$$
which is the distance from the crack tip. To determine the theoretical size of the plastic zone (*r_t_*) near the crack tip, set the stress *σ* equal to the material’s yield strength *σ_y_* and solve for the distance *r* from the crack tip. Using the stress field equation as follows:(7)$$r_{t}=\frac{1}{2\pi }{\left(\frac{K_{app}}{\sigma_{ty}}\right)}^{2}$$
where *K_app_* represents the stress intensity due to the applied stress, and *σ_ty_* is the material’s tensile yield strength. For the actual plastic zone size to match the theoretical prediction, the stresses within the plastic zone must exceed the yield strength. However, because the material cannot sustain significantly higher stresses, the stress near the crack tip redistributes to regions further away. As a result, the actual plastic zone is approximately twice the theoretical size, or about (*2r_t_*). Figure 2 shows the theoretical elastic stress, plastic zone size, redistributed stresses, and the realistic estimate of the plastic zone size. When yielding occurs, stresses redistribute to maintain equilibrium. In the plastic zone, replacing the elastic stress distribution with the constant yield stress disrupts the equilibrium along the y-direction.

The cross-hatched area in Figure 2 shows the forces that the elastic material could support, but the elastic–plastic material cannot support since the stresses are capped at the yield. To balance these forces, the plastic zone expands. A simple force balance assumes the force carried by the elastic stress distribution remains the same before and after yielding, providing a second-order estimation as follows:(8)$$\sigma_{ty}r_{p}=\int_{0}^{rt}\sigma_{yy}dr=\int_{0}^{rt}\frac{K_{I}}{\sqrt{2\pi r}}dr$$

Hence, this results in Equation (9):(9)$$r_{p}=\frac{1}{\pi }{\left(\frac{K_{I}}{\sigma_{ty}}\right)}^{2}$$

The governing equation based on the fracture mechanics model is used to evaluate the fracture mechanics variables. The methodology for the following equations and the sequence of analysis were taken according to the literature [40,41,42,43,44,45,46,47]. The initial parameter to compute is the linear elastic fracture mechanics (LEFM), employing the concept of the stress intensity factor. Typically, the plane–strain fracture toughness, (*K_IC_*), is chosen as the critical stress intensity value for the design and analysis. Subsequently, the factor of safety is determined as follows:(10)$${FS}_{LEFM}=\frac{K_{IC}}{K_{app}}$$

In a hydraulic fracture assessment, the dam’s behavior is often modeled as a plate, requiring verification of the LEFM applicability. For the LEFM to be valid, the plastic zone must be small relative to the structure and the crack geometry. If the plastic zone extends too close to the boundaries, it can cause significant yielding and deformation. The plastic zone forms ahead of the crack tip, and the crack tip should be positioned at least a distance (*d_LEFM_*) from any boundary, which in plane stress conditions is about four times the plastic zone size. This is demonstrated by Equation (11):(11)$$d_{LEFM}=\frac{4}{\pi }{\left(\frac{K_{app}}{\sigma_{ty}}\right)}^{2}$$

If the LEFM is not applicable, an elastic–plastic analysis based on the failure assessment diagram (FAD) is required to account for the plasticity effects around the crack [42]. The applicability of the LEFM was assessed based on the relative size of the plastic zone at the crack tip to the uncracked ligament and plate dimensions. When the calculated plastic zone *r_t_* exceeded approximately 10% of the remaining ligament width, the LEFM was deemed invalid. A FAD is a graphical method used to evaluate the integrity and safety of structures with cracks or defects, combining both the fracture and the plastic collapse failure modes. By plotting the material’s response on the FAD, engineers can assess whether the structure is safe or at risk. The FAD helps determine design acceptability by comparing the applied loading conditions with the material properties, like fracture toughness and yield strength. Both the stress ratio (*S_r_*) and the stress intensity ratio (*K_r_*) must be calculated to assess the design’s safety under specific load cases, as follows:(12)$$S_{r}=\frac{\sigma_{app}}{\sigma_{ty}}$$(13)$$K_{r}=\frac{K_{app}}{K_{IC}}$$
where *σ_app_* represents the applied stress, *K_app_* is the stress intensity factor at the applied stress, *σ_ty_* is the material’s tensile yield strength, and *K_IC_* is the material’s plane–strain fracture toughness. To determine the factor of safety, a line (referred to as the load line) is drawn from the origin through the design point, continuing until it intersects the FAD failure locus, see Figure 3. The factor of safety is defined as the ratio of the length of the load line between the origin and the failure point to the length of the load line between the origin and the design point as shown in the following equation. The FAD failure locus is expressed as follows:(14)$$K_{r.f}={\left[\frac{E\varepsilon_{ref}}{S_{r}\sigma_{ty}}+\frac{S_{r}^{3}\sigma_{ty}}{2E\;\varepsilon_{ref}}\right]}^{-0.5}$$
where *E* is the material’s elastic modulus, *σ_ty_* is the material’s tensile yield strength, and *S_r_* is the stress ratio defined earlier. The value $\varepsilon_{ref}$ represents the true strain corresponding to the stress *S_r_.σ_ty_*, which can be calculated using the Ramberg–Osgood equation. One of the key advantages of the FAD approach is its ability to account for material plasticity while still utilizing linear elastic stress intensities, offering simplicity compared to other elastic–plastic methods. A failure assessment diagram was used to assign the safety factor to represent the safety monitoring index, due to the limited applicability of the LEFM according to the chosen plates’ dimensions. The severity of a cracked component is characterized by the stress intensity factor *K*, with failure occurring when *K = K_c_*.

On the other hand, for the calculation of the critical crack length, the residual strength of a cracked component is given by Equation (15):(15)$$\sigma_{c}=\frac{K_{c}}{Y\sqrt{\pi a}}$$
where *Y* represents a geometry correction factor. The stress *σ* denotes the gross stress on the section where the crack length *a* is defined, with the residual strength referring to a net section condition. Under plane strain conditions, *Kc = K_IC_*, the crack size corresponding to this stress is known as the critical crack size. Solving the critical crack size analytically is challenging because *Y*(*a*) is typically a complex function of the crack length and the component geometry [42]. However, this can be determined numerically through iteration. When *Y* changes slowly with the crack length, for example, for a small crack in a wide panel, an approximate value of *Y* can be used to estimate the critical crack size that a component can withstand under a given load using the following equation:(16)$$a_{c}=\frac{1}{\pi }{\left(\frac{K_{c}}{Y\left(a_{c}/W\right)\sigma }\right)}^{2}$$

The two equations above form the foundation of the fracture mechanics-based design methodologies [42]. It is important to note that Equation (15) is valid only when the linear elastic fracture mechanics applies specifically, when the net section stress is well below the material’s yield stress. If the stress level approaches the yield stress, the component may fail by plastic collapse rather than fracture. For a center-cracked panel with a finite width *W*, the maximum load-carrying capacity is limited by the plastic collapse strength, where the stress across the entire section reaches or exceeds the material’s yield strength or the ultimate tensile strength. The nominal stress at collapse can be derived straightforwardly through Equation (17):(17)$$\sigma_{pc}=\frac{W-2a}{W}\sigma_{ty}$$

When this occurs, fracture will occur regardless of the fracture toughness since the plastic deformation becomes unbounded. Thus, plastic collapse and brittle fracture are the two potential failure modes. A collapse will occur if the fracture stress *σ_c_* is greater than the stress that causes failure by collapse. Consequently, for *σ_c_* and *σ_pc_*, the real residual strength is the lowest. In the event of a center cracked panel, a plastic collapse failure would occur in three different scenarios: The width *W* is very small, the crack is very small, and the toughness is very high. Figure 4 shows a sketch for the potential failure modes. The two curves’ intersection is given in Equation (18):(18)$$\frac{W-2a}{W}\sigma_{ty}>\frac{K_{c}}{\sqrt{\pi a}\sqrt{sec(\pi a/W}}$$

A comprehensive investigation of the concrete gravity dam’s resistance to hydraulic fracture was carried out using a mathematical fracture evaluation model, considering factors like the ultimate overloading coefficient and the crack depth. The ultimate overload coefficient was assessed through the fracture mechanics splitting criterion to express the dam’s resistance and to determine the crack depth. A strength-based criterion was applied, expecting crack expansion if the tensile stress at the fracture tip exceeded the concrete’s allowable tensile strength [20]. The three main stress modes include (I) the opening mode, (II) the in-plane shear mode, and (III) the out-of-plane tearing mode (see Figure 5). The dam’s resistance was examined under two failure modes: open-type failure and in-plane shear failure. The Guxian dam’s safety was theoretically assessed using the model, and the results were compared across different failure modes, with the splitting criterion considering both uplift pressure and its absence.

## 3. Analysis Procedures

### 3.1. Safety Monitoring Index with and Without Considering the Weak Layers

The safety index for the Guxian dam was evaluated using a 2D model with a FEM analysis conducted in ABAQUS 6.14. The assessment considered both scenarios with and without the weak layers present in the RCC dam. This study investigated the impact of weak interfaces on the long-term safety of the Guxian high concrete gravity dam. To track the safety variations during the dam’s progressive failure, the strength reduction method was applied as an indicator to analyze dam displacement and the connectivity of yield zones. Material strength deterioration is identified as one of the most critical factors affecting dam stability. Therefore, the strength reduction method offers an effective approach to understanding the failure process of high concrete gravity dams under this adverse condition. The factor of safety is calculated in the normal case of loading. The material properties used in the FEM simulations are listed in Table 2.

The boundary condition of the dam model is as follows: the upstream and downstream limits of the foundation were subjected to level constraints, ensuring that these areas remained stable and aligned with the expected ground levels. The bottom boundary of the foundation was fixed to prevent any vertical movement, providing a stable base for the dam. This fixed constraint effectively modeled the interaction between the dam and its foundation, allowing for the accurate representation of stress and stability conditions. On the other hand, the perimeter of the dam body was set as a free boundary, allowing the dam’s surface to deform naturally without any restrictive constraints.

The distribution of material properties throughout the Guxian dam body is illustrated in Figure 6a. While Figure 6b presents a 2D model of the Guxian dam. Level constraints were applied to the upstream and downstream boundaries of the foundation, while the bottom boundary of the foundation was assigned a fixed constraint. The boundaries of the dam body were designated as free boundaries. Four monitoring points at the dam cross section were determined as the fixed points of dam displacement measurements, and the elevations of these points are 631 m (dam crest), 525 m, 480 m, and 465 m, as shown in Figure 7. The displacement of the dam body was recorded at these points at different strength reduction factors. The assessment indicators of the dam displacement and the connectivity of yield zones were used to express the safety of the Guxian High Dam in the progressive failure procedure considering the weak layers. In order to simplify the analysis process and to prevent the complexity of the FEM model, the weak interface between the layers of the RCC dam was taken every three meters in the current model with a thickness of 1 m, according to [48].

Usually, the thickness of the RCC dam layers vary from a few centimeters to extend to some meters, according to the design and construction conditions, based on standard construction practices and supported by guidelines, such as those from [48], which specify that individual roller-compacted concrete (RCC) lifts can reach up to 3 m in thickness. To reasonably simulate the influence of these weak interlayers within a finite element model (without introducing excessive complexity) a simplified assumption was adopted. Specifically, a 50 cm thickness was assigned to represent the weakened zone from the lower surface of an upper lift, and another 50 cm from the upper surface of a lower lift, resulting in a total weak layer thickness of 1 m at each interface. That assumption was chosen after some trials (up to seven models) to reach the reasonable simulation representing the problem through these modeling.

### 3.2. Comparative Analysis Between Different Crack Locations Influence of Different Crack Locations on the Dam Safety

The fracture evaluation mechanical model was used to comprehensively study the hydraulic fracture and crack propagation behavior for high dams in case of different crack locations. A center-through crack and a single edge-through crack in the plates has been chosen for the comparison due to their high possibility of occurrence for concrete dams. In high concrete dams, a semi-infinite surface crack (edge crack) originates at the surface, typically near the edges or high-stress areas, and extends through the entire thickness of the dam. A center-through crack is located internally, running symmetrically through the dam’s thickness. For the comparison completion, two similar plates were used for both cracks identified with the same material properties for the Guxian dam, for the schematic sketch of the two cracks see Figure 8, while the material properties are listed in Table 3 representing the Guxian dam material properties used for the plates. The full dimensions for the plates under investigation are listed in Table 4 [49]. The safety factor, evaluated using the failure assessment diagram (FAD), represents the first safety index related to crack propagation. The critical crack depth serves as the second safety index evaluated through the residual strength curve. Together, these two indexes assess the dam instability and failure risk due to crack propagation.

### 3.3. Comprehensive Safety Analysis of the Guxian Dam Safety Resistance to Hydraulic Fracture at Different Modes Failure Investigation of Guxian Dam Resistance to Hydraulic Fracture 

In this part of the research, a comparison between the first and second mode of fracture mechanics failure is discussed through evaluating the ultimate overloading coefficient in both modes to determine the dam resistance during these two types of failure. Table 5 represents the principal stresses for the crack Modes I and II, which are considered in the current analysis.

Here (*r*, *α*) represent the polar coordinates centered at the crack tip, the variable *r* denotes the distance from an arbitrary point to the crack tip, while *α* is the angle between the *X*-axis and this point. See Figure 9 for an illustration. These coordinates describe how the stresses vary with both the distance from the crack tip and the angular displacement from the *X*-axis.

Since displacements and stresses are linearly related to the stress intensity factor, the fracture problems can be addressed using the superposition principle. This approach, supported by the handbooks, is a key tool for applying fracture mechanics to practical problems, such as mixed-mode loading, crack interaction analysis, and combined stress field evaluations. The principle states that the stresses caused by different loads can be added together, but it is only valid when the structure is subjected to different loads of the same mode. For example, when a component fails under combined tension and bending, the crack tip stresses can be calculated using the following equation:(19)$$\sigma_{ij}=\frac{K_{I,\;II}^{tension}}{\sqrt{2\pi r}}f_{ij}\left(\alpha \right)+\frac{K_{I,II}^{bending}}{\sqrt{2\pi r}}f_{ij}\left(\alpha \right)$$

Because the angular function *f_ij_*(*α*) is the same for the same fracture mode, the above equation can be rewritten as follows:(20)$$\sigma_{ij}=\frac{K_{I,II}^{total}}{\sqrt{2\pi r}}f_{ij}\left(\alpha \right)$$

For Mode I, the stress intensity is as follows:(21)$$K_{I}=Y\;\sigma \sqrt{\pi a}$$

For Mode II, the stress intensity is as follows:(22)$$K_{II}=Y\;\tau \sqrt{\pi a}$$

In this context, *τ* represents the applied shear stress along the base of the dam or within the potential failure planes caused by the horizontal sliding forces from water pressure. Due to the large volume of high concrete gravity dams, cracks on the upstream surface are often approximated as semi-infinite surface cracks. Horizontal cracks are assumed to form at the dam heel, with crack propagation influenced by high water pressure within the crack, and vertical stress at the heel in the uncracked condition. This results in Mode I (tensile opening) crack behavior. For Mode II (in-plane shear), shear stresses on the horizontal planes are considered in the calculation. The stress intensity factor at the crack tip is determined by superposition of the forces of water pressure and stress. Thus, the stress intensity factor for a horizontal crack at the dam heel of a high concrete gravity dam can be calculated. The sequence for calculating these stresses, using the gravity method, is outlined below for both modes, as follows:(23)$$\tau_{u}=\left(p_{w}-\sigma_{zu}\right)\tan\emptyset_{u}$$(24)$$\tau_{d}=\sigma_{zd}\tan\emptyset_{d}$$
where $\sigma_{zu}$ is the normal stresses on horizontal planes on the upstream face, and $\sigma_{zd}$ is the normal stress on the downstream face. According to the requirements in the code for the design of gravity dams [50], with ∑M counterclockwise as positive and ∑P gravity direction as positive, the normal stresses can be calculated at the dam heel as follows:(25)$$\sigma_{z}=\frac{\sum P}{B}+\frac{6\sum M}{B^{2}}$$

Thus, the total stress intensity factor is as follows:(26)$$K=\sqrt{\pi a}\left[\frac{P}{BW}f_{P}\left(\frac{a}{w}\right)+\frac{6M}{BW^{2}}f_{M}\left(\frac{a}{W}\right)\right]$$

For a ratio *a/w* = 0.2, the stress intensity factor can be expressed as follows:(27)$$K=\sqrt{\pi a}\left(1.21\frac{P}{BW}+1.055\frac{6M}{BW^{2}}\right)$$

The forces acting on the dam and the location of the horizontal crack are illustrated in Figure 10. While *P*_1_ is the water pressure at the dam heel plus the vertical tensile stress, subtract the vertical compressive stress at the same position where it is seamless; *P*_2_ is equal to the water pressure at the dam heel plus the vertical tensile stress, or subtract the vertical compressive stress at the crack tip. These two forces can be calculated as shown in Equations (28) and (29).(28)$$P_{1}=\gamma H_{1}-\left[\frac{\sum W}{B}+\frac{6\sum M}{B^{2}}\right]$$(29)$$P_{2}=\alpha_{1}\gamma H_{1}-\left[\frac{\sum W}{B}+\frac{\sum M\left(\frac{B}{2}-a\right)}{\frac{B^{3}}{12}}\right]$$

Since *P*_2_ is located at the crack tip, the reduction in water pressure within the crack becomes significant. The water pressure at the crack is given by *α*_1_*γH*_1_, where *α*_1_ is the attenuation coefficient of concrete permeability, representing the reduction in water head from the crack mouth to the crack tip [51]. Consequently, the stress at this location can be calculated using the previously established equations, with the convention that the resultant force due to gravity is positive, and the counterclockwise movements are considered positive. For *P*_1_, located at the crack opening, there is no reduction in water pressure. Therefore, the water pressure at this point is expressed as *γH*_1_. The stress at *P*_1_ can be determined using the corresponding equation, again with the assumption that the resultant force due to gravity is positive, and the counterclockwise movements are treated as positive.(30)$$P_{1}-P_{2}=\left(1-\alpha_{1}\right)\gamma_{w}H_{1}\frac{12.a\sum M}{B^{3}}$$

The fracture toughness (*K_IC_*) of concrete in high concrete gravity dams is determined using an empirical equation. According to [52,53], the empirical formula for concrete fracture toughness is expressed as follows:(31)$$K_{IC}=1.9\beta f_{t}$$

Here *β* ranges from 0.2 to 0.3 and is typically taken as 0.22, and *f_t_* represents the axial tensile strength of the concrete material, measured in Pascals (Pa). By taking Equations (27)–(31) into the *K* = *K*_IC_ criterion, the critical crack propagation equation at the dam heel of the high concrete gravity dam can be obtained as follows:(32)$$\left[{0.66\alpha }_{1}\gamma H_{1}+0.439\gamma H_{1}-1.1\left[\frac{\sum W}{B}+\frac{6\sum M}{B^{2}}\right]+\frac{7.932\;a\sum M}{B^{3}}\right]\left(\sqrt{\pi a}\right)=0.418f_{t}$$

In this analysis, *H*, *B*, and *a* represent the water head at the dam heel, the width of the dam section, and the equivalent crack depth, respectively, all in meters, *ΣW* denotes the vertical component of the resultant force in Mode I and the shear forces in Mode II, while *N* and *ΣM* represent the resultant movement in the absence of a crack, in N·m. The crack depth and ultimate overloading coefficient were evaluated under two scenarios: with and without considering uplift pressure. For the uplift pressure case, two approaches were used: one assuming full uplift and another with reduced uplift (referred to as the first and second methods). The parameters of the Guxian dam used in the analysis are listed in Table 6.

The modeling of the uplift pressure in the dam body was approached in two distinct ways. In the first approach, the uplift pressure on the cracked section was neglected, and only the splitting water pressure within the crack was taken into account. In the second approach, the influence of the uplift pressure along the cracked section was explicitly considered. Two variations of the uplift pressure treatment were adopted. One method ignored the reduction effect from the anti-seepage curtain and approximated the uplift distribution using an inclined linear form. The other method incorporated the reduction effect of the curtain, providing a more realistic representation of the field conditions.

## 4. Results and Discussion

### 4.1. Safety Monitoring Index Considering Weak Layers

Dam displacement and connectivity of the plastic yield zones were used as indicators of the dam progressive failure with and without considering the weak layers of the RCC dam. It has been noticed that the mode of failure has changed after considering the weak interfaces of the RCC dams, and the safety factor has dropped noticeably from 3.8 to 3.4 in the case of considering these weak layers (see Figure 11a and Figure 12a). The ultimate strength reduction ratio is considered the safety factor (or strength reserve coefficient) and is obtained when the penetrating plastic zone appears and is fully connected, and the structure is destabilized through the gradual reduction of the shear strengths of the soft interlayers. Due to the existence of the stream direction for the horizontal weak interlayer, the yield zones first appeared at the dam toe. When the reduction parameter reaches values of more than 3.4, in the case of considering the weak layers (see Figure 12a), the connectivity of the plastic zones is formed at the dam toe, namely the potential downstream sliding surface is developed. With the continual increments of the reduction parameter, the yield zones rapidly extend towards the upstream direction. It can be noticed that, when the strength reduction parameter is less than 3.4, the displacement of the selected four typical points remains almost constant (see Figure 12b), with only a slight increase noticed, and the plastic yield zones only cover a smaller area of the dam cross section. The semi-constant line in the relationship between the dam displacement and the reduction factor indicates that the dam is still stable. When dam concrete material deformation takes place in a larger area, and in parallel, the dam displacement gradually increases. The explanations may be attributed to the fact that the connectivity of the plastic yield zones in the dam body was not developed completely before the strength reduction factor reached 3.4. When the strength reduction exceeds 3.4, the plastic yield zones largely form in the dam body, which leads to a rapid increase in the deformation at the upper part of the dam that represents failure. The same trend can be noticed for the case of not considering the weak layers, but the dam shows more resistance, and the failure happens at the higher value of safety factor equals 3.8; see Figure 11. That confirms the negative impact of the weak layers on the dam safety. The results clearly showed that the presence of weak layers significantly reduces the dam’s safety. Specifically, the safety factor decreased from 3.8 to 3.4 when the weak layers were considered, corresponding to the point of maximum plastic strain and full connectivity of the plastic zones in the dam cross section. This reduction highlights the destabilizing effect of construction-induced interfaces. These findings are consistent with those reported in [34], which demonstrated that the presence and positioning of weak layers significantly influence the damage characteristics and failure modes in concrete dams. The observed agreement further validates the importance of incorporating weak layer effects into dam safety evaluations.

Table 7 illustrates the safety monitoring index for the Guxian dam in both cases with and without considering the weak layers.

Including the weak layers of the Guxian RCC dam has decreased the dam safety factor, therefore the weak layers influence should be highly considered during the stage of design due to its negative effect on the dam safety during the long-term operation. The safety index, when considering the presence of the weak layers, is lower compared to the scenario where these layers are not considered. This indicates an increased safety risk due to the existence of the weak layers, with noticeable variations in the safety index. The results show that displacement, which signifies progressive failure, was significantly smaller in the lower part of the dam cross-section reduced by 20% compared to the case where the weak layers were not accounted for. The influence of the weak layers in the lower cross-section is evident, as this area contains larger interfaces between the layers than the upper part. This increased interface area heightens the potential for material deterioration, which may lead to failure. These findings suggest a heightened risk associated with the weak layers in the lower cross-section, making it a critical region for potential instability and failure.

### 4.2. Comparative Analysis Between Different Crack Locations

In this part of the analysis, the safety monitoring index has been evaluated for the Guxian dam mathematically using the fracture evaluation mechanical model. A plate with different dimensions has been used to represent the Guxian dam material properties and the crack propagation process. Two different safety indexes have been calculated, including the safety factors using the failure assessment diagram and the critical crack depth using the residual strength diagram. Both safety indexes represent the dam failure points during crack propagation. The risk due to crack propagation can be explained through a comparison between two different crack locations and different plate dimensions. The safety monitoring index has been assigned through calculating the safety factors as shown in Figure 12 and Figure 13. The failure assessment diagram (FAD) was used to assess the dam’s resistance to cracks and to determine the safety factor. The FAD, which combines plastic collapse and brittle fracture failure criteria into a single diagram, helps evaluate whether a cracked component will fail under a given load. The factor of safety is the ratio of the length of the load line between the origin and the failure point, and the length of the load line between the origin and the design point. Figure 13 shows the FAD for the plates analyzed in the case of a center-through crack.

For a center-through crack, factors like symmetrical stress distribution around the crack enhance the structural stability and reduce the influence of boundary proximity, unless in small plates. It also has more uniform load distribution, lowering local stress intensity. By contrast, the single-edge crack, shown in Figure 14, has asymmetrical stress concentration, leading to higher localized stress near the crack tip. This type of crack is more prone to propagation due to surface loading or environmental factors like water pressure, making it more susceptible to hydraulic fracture in high concrete dams.

Single-edge cracks in high concrete dams pose a greater threat to structural integrity than center-through cracks due to asymmetric stress, concentration, and sensitivity to boundary effects. These cracks are more likely to propagate under applied loads or environmental conditions, especially under tensile or shear forces. Proximity to structural boundaries, such as dam heels or surfaces, allows for the plastic zone to interact with the edge, reducing the support and increasing the risk of unstable propagation. Surface cracks experience direct water pressure and external loads, enhancing crack extension potential, while center-through cracks are less affected by surface loads. For the critical crack length, see Figure 15 and Figure 16 for the residual strength curve for both crack locations. The part’s strength as a function of crack size is indicated on the residual strength curve. To assess the acceptability of a design, plot the design point (a,σ_app_), for instance, where a is the crack length and σ_app_ is the applied combined stress. From this point, draw a vertical line upward to the residual strength curve. This intersection indicates the failure point if the crack size remains constant, while the stress is increased to its critical (failure) value. Next, draw a horizontal line from the design point to the residual strength curve. This intersection shows the failure point if the stress is held constant, while the crack size is increased to its critical (failure) value. The safety monitoring indexes are listed in Table 8.

The safety factors for both center-through and single-edge cracks remain stable across the different plate’s dimensions, with minor variations between the crack locations. The safety factors for both cases are safer, while the single-edge crack shows a lower safety factor compared with the center-through crack. As the crack length increases, the critical crack depth (*a_cr_*) decreases. For both crack types, a clear inverse relationship is observed, with the critical crack length declining as the safety factor approaches its minimum threshold. For example, Plate 1 (I) shows a critical crack length of 0.2157 m for the center-through crack and 0.1176 m for the single-edge crack, while Plate 4 (IV) demonstrates a much lower critical crack length (0.04 m for center-through cracks and 0.01795 m for single-edge cracks). In general, the safety factors indicate a higher degree of stability for center-through cracks, whereas single-edge cracks consistently yield lower safety factors, suggesting a greater vulnerability to failure. The findings indicated that single-edge cracks impose a significantly higher risk than center-through cracks, with a 10% lower safety factor and a 40% shorter critical crack length. These findings have shown a good agreement with the study presented by [54], where the author carried out a comparison between different crack locations numerically and theoretically including edge and center cracks, and illustrated that edge cracks have higher SIF.

The results highlight that, as crack propagation progresses (larger crack lengths), the structural integrity of the plates becomes increasingly compromised, as indicated by both a reduction in the safety factor and a decrease in the critical crack length, particularly for single-edge crack. This reinforces the importance of monitoring crack growth to predict the potential failure and to ensure the continued safety of the structure. However, both crack locations have a higher risk for the dam safety, and single-edge cracks show higher potential hazard for the dam and the occurrence of hydraulic fracture phenomena.

### 4.3. Comprehensive Safety Analysis of the Guxian Dam Safety Resistance to Hydraulic Fracture at Different Modes Failure

The analysis calculated the crack depth by varying the upstream water level and using the overload coefficient to quantify the hydraulic fracture resistance. The results showed that increasing the overload coefficient deteriorated stress conditions at the dam heel, causing non-linear crack depth reductions. Mode I failure was more critical due to the lower ultimate overload coefficient, indicating weaker resistance to hydraulic fracturing as tensile stresses are more sensitive to hydraulic pressure. Mode II showed higher resistance, as shear stresses are less affected by hydraulic conditions. Including uplift pressure significantly reduced the ultimate overload coefficient, especially in Mode I, where cracks were initiated earlier due to the rapid tensile stress concentration at the crack tip. Mode I, driven by tensile stresses perpendicular to the crack plane, causes faster crack propagation due to the concrete’s low tensile strength. By contrast, Mode II involves shear stresses parallel with the crack plane, which are better resisted due to higher shear strength and internal cohesion. Cracks in Mode II follow an irregular path, dissipating stress concentrations and requiring more energy to propagate. The increased sensitivity of Mode I to uplift pressure underscores the need for design measures to reduce the uplift pressure and to mitigate hydraulic fracture risks. Figure 17 compares both failure modes.

The analysis results indicate that Mode I (tensile failure) poses a higher risk to the Guxian dam compared to Mode II (shear failure). The ultimate overload coefficient without considering uplift, which represents the dam’s resistance to hydraulic fracture, is lower in Mode I (1.27) than in Mode II (1.32). This confirms that the dam has more resistance to hydraulic fracture in Mode II, supporting the theory that tensile stresses, which dominate Mode I, lead to a weaker overall resistance. The higher fracture energy observed in Mode II compared to Mode I can be attributed to the fundamental mechanisms of crack propagation in concrete. Mode I, the tensile opening mode, typically initiates a relatively clean crack (the crack tends to open directly and smoothly) with minimal resistance once the tensile strength is exceeded. By contrast, Mode II involves in-plane shear displacement along the crack surface, which engages a wider fracture process zone. Within this zone, several energy-absorbing mechanisms become active, including the formation of inclined tensile microcracks, sliding, and friction between aggregate particles, and the interlock of rough crack surfaces. These mechanisms significantly increase the energy required for the crack to propagate in shear. Aggregate interlock plays a particularly crucial role, as the asperities along the crack faces generate frictional resistance that must be overcome for sliding to occur. Additionally, the presence of residual cohesion across the crack plane further contributes to Mode II’s higher fracture toughness. These factors collectively result in fracture energies for Mode II that are typically 20 to 25 times greater than those for Mode I, as documented in the studies by [36,37]. This theoretical understanding aligns with the current study’s numerical results, where the ultimate overload coefficient in Mode II was higher, indicating greater resistance to hydraulic fracture in shear-dominated failure scenarios. The lower ultimate overload coefficient for Mode I reflects the increased vulnerability of the dam to crack initiation and propagation under tensile stress, making Mode I the more critical failure mode. Both modes of failure can happen together, but usually Mode I has the higher possibility to occur first, and in different locations, and could expose more risk to the dam safety.

### 4.4. Limitations

This study uses a simplified representation of the dam structure as a plate to model the crack behavior defining the plates with the material properties of the real dam. This assumption may not accurately reflect the complex geometry of real dams, which may have irregular shapes, varying thicknesses, and additional structural features, such as joints or reinforcements that affect crack propagation. The choice of plate dimensions (width and thickness) plays a significant role in the stress intensity factor calculation. Real-world dams may exhibit variations in these dimensions, leading to different stress distributions that could affect crack initiation and growth. The limited application of a uniform plate geometry may reduce the accuracy of the results when applied to actual dam structures. This study uses a 2D model to analyze dam safety and crack propagation for simplicity and computational efficiency. While suitable for cross-sectional evaluations, this approach does not capture 3D effects, such as out-of-plane stresses, or transverse confinement. As a result, the findings may not fully represent the behavior of the entire dam structure. Future studies should consider 3D modeling to better capture spatial interactions and to incorporate dynamic loading for more realistic hydraulic fracture simulations.

## 5. Conclusions

This study established safety monitoring indexes for the Guxian high concrete RCC gravity dam, considering weak layers and crack propagation numerically and mathematically, thereby highlighting the possible risk of hydraulic fracture on the high concrete gravity dams, and showing the necessity of including these phenomena into the design regulation for high concrete gravity dams. Additionally, the dam’s resistance to hydraulic fracture was investigated under two distinct failure modes, Mode I and Mode II. Based on these investigations, the following conclusions have been drawn:A numerical model was used to evaluate the safety monitoring index of the Guxian RCC dam, considering the impact of weak layers. The results showed a significant reduction in dam safety, particularly in the lower part of the cross-section, where the safety index decreased by 20%. These weak layers contributed to the formation of extensive plastic zones, reducing the dam’s overall stability. By contrast, the dam crest was less affected, as the weak layers had smaller interface areas in this region, leading to only a 3% reduction in the safety index.Two different crack locations were analyzed to assess the safety index of the Guxian dam. The findings revealed that a single-edge crack poses a significantly greater threat to dam safety compared to a center-through crack. Specifically, the safety factor derived from the FAD decreased by 10% for the single-edge crack compared with the center-through crack. Additionally, the critical crack length for this crack type was 40% lower than that of the center-through crack, exposing the dam to a substantially higher risk.The resistance of the Guxian dam to hydraulic fracture was assessed using a hydraulic fracture mechanical model, calculating the ultimate overload coefficient for two failure modes, Mode I (opening mode) and Mode II (in-plane shear mode). The analysis showed that the dam’s resistance was lower in Mode I due to the concrete’s weaker resistance to tensile stress compared to shear stress. Mode II fracture energy is generally higher than Mode I, as it accounts for the formation of inclined tensile microcracks within the fracture process zone and the energy needed to overcome shear resistance from aggregate interlock and surface asperities behind the crack tip. The ultimate overload coefficient for Mode I was 5% lower than for Mode II. Although both failure modes pose a risk of hydraulic fracture, Mode I presents a greater threat and should be prioritized in safety evaluations.Crack propagation and the presence of weak layers in RCC dams are critical engineering challenges that should be explicitly addressed in design regulations and considered during the design stages. Both factors significantly impact dam safety, particularly during long-term operation.

## Figures and Tables

**Figure 1 materials-18-02893-f001:**
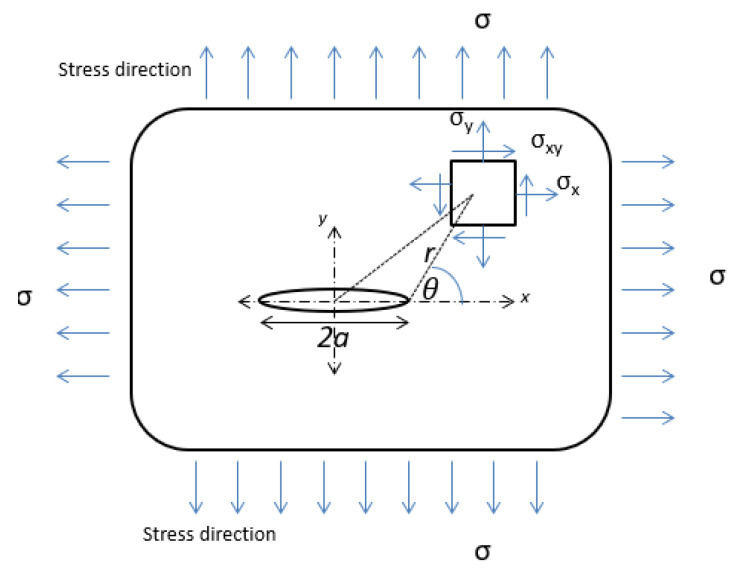
A crack of length 2a in an infinite plate [20].

**Figure 2 materials-18-02893-f002:**
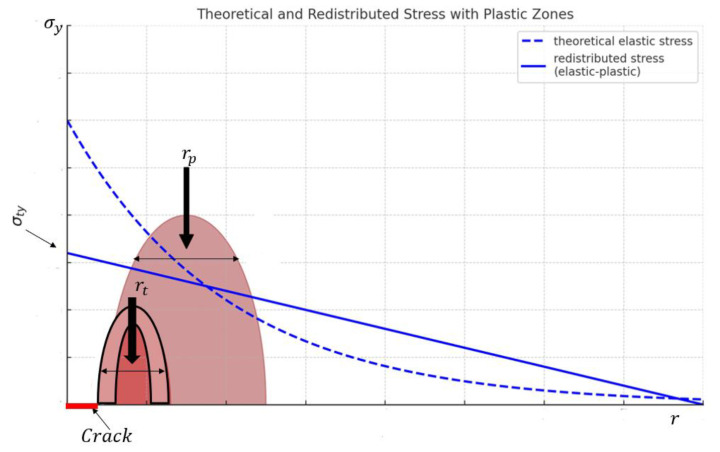
Illustration of theoretical elastic stress and plastic zone size [20].

**Figure 3 materials-18-02893-f003:**
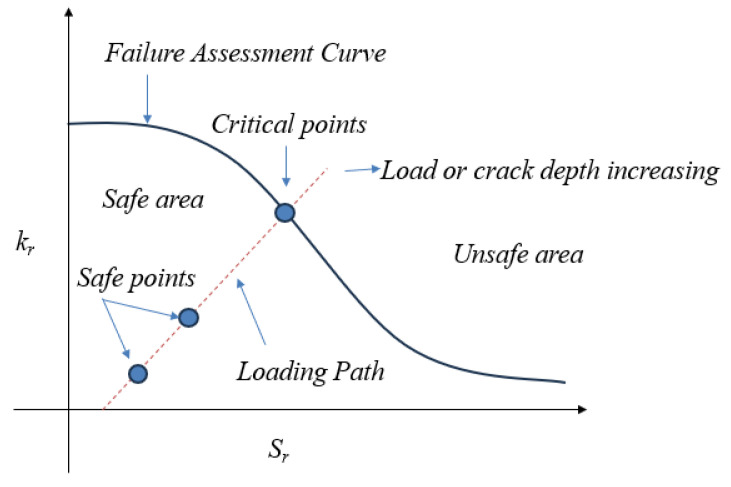
Failure assessment diagram (FAD).

**Figure 4 materials-18-02893-f004:**
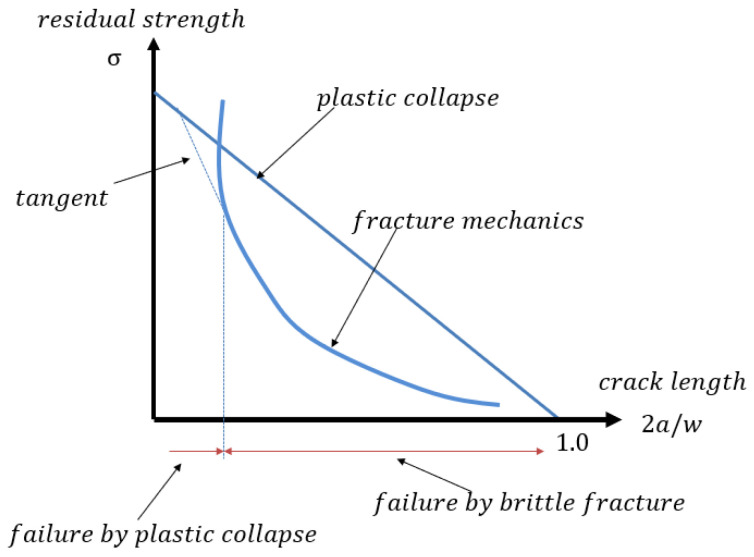
Schematic graph for two modes of failure fracture and collapse [20].

**Figure 5 materials-18-02893-f005:**
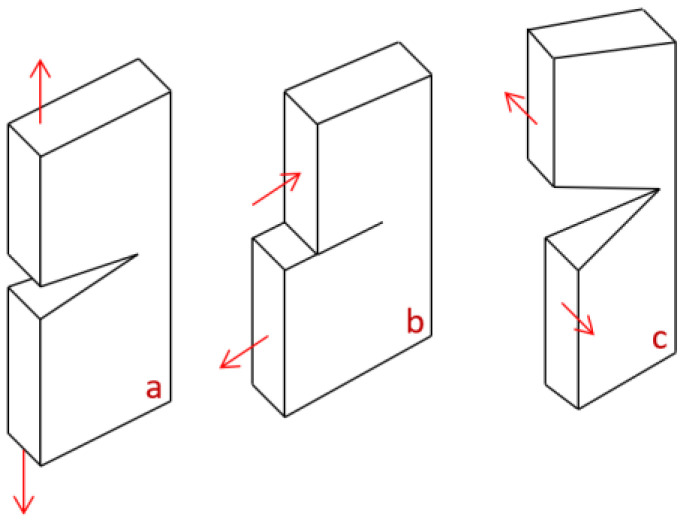
The three basic modes of crack extension: (**a**) opening mode, (**b**) sliding mode (in plane shear), and (**c**) tearing mode (out of plan shear).

**Figure 6 materials-18-02893-f006:**
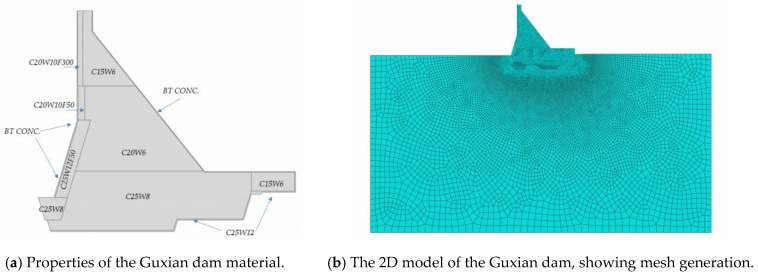
FEM model.

**Figure 7 materials-18-02893-f007:**
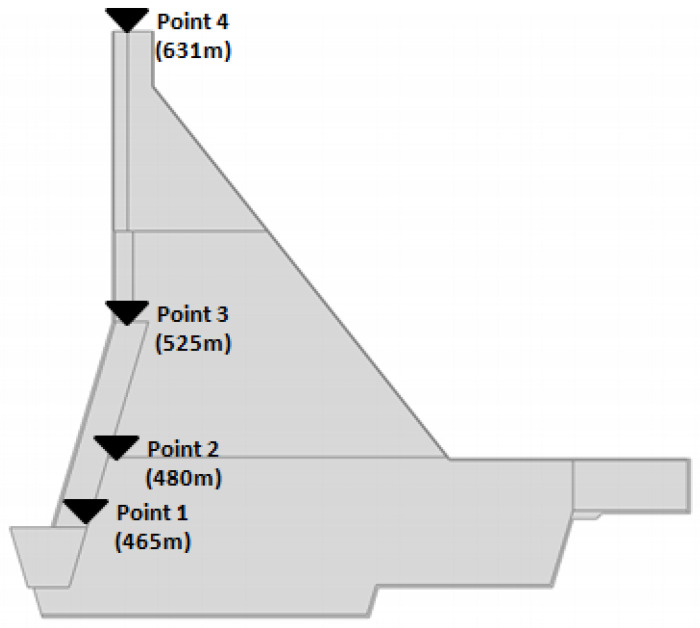
Typical displacement points along the dam cross section.

**Figure 8 materials-18-02893-f008:**
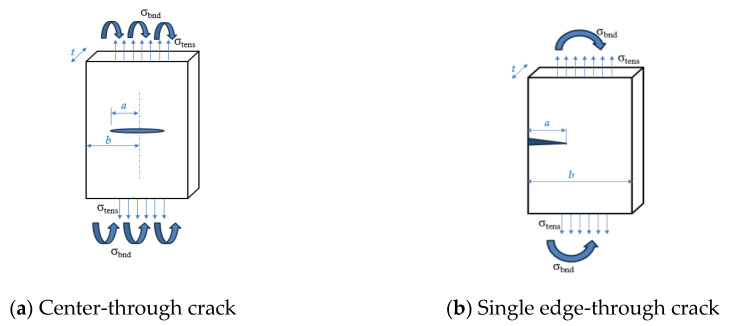
Different crack locations.

**Figure 9 materials-18-02893-f009:**
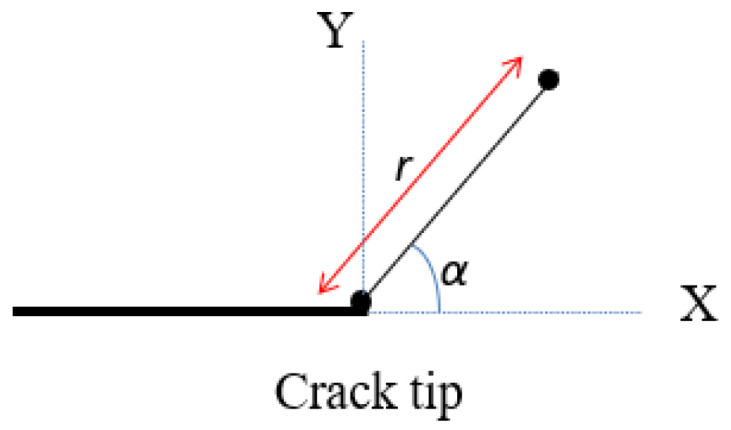
Polar coordinates (r, α) representing the distance and angle from the crack tip to the calculation point.

**Figure 10 materials-18-02893-f010:**
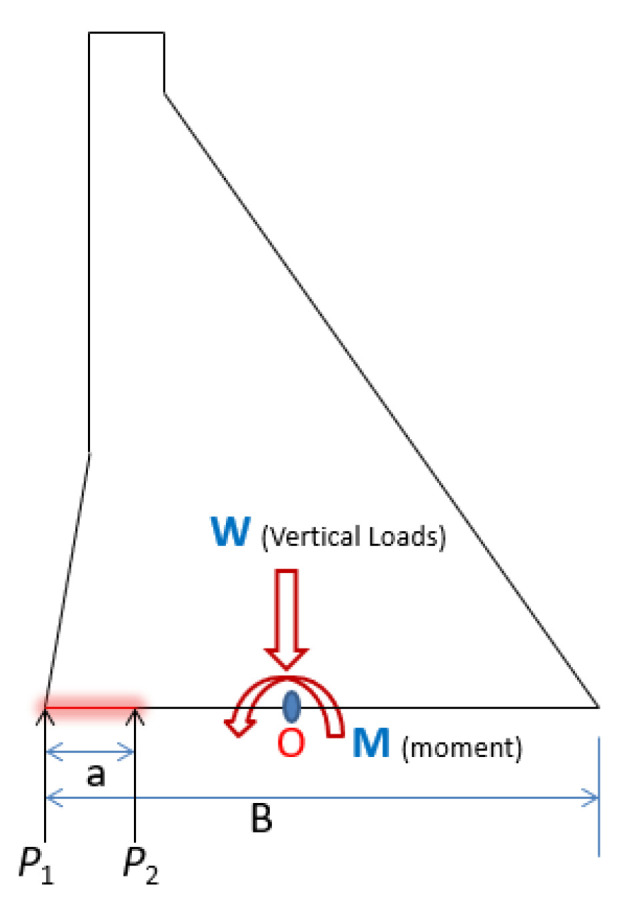
Force diagram for the dam body’s horizontal cracks.

**Figure 11 materials-18-02893-f011:**
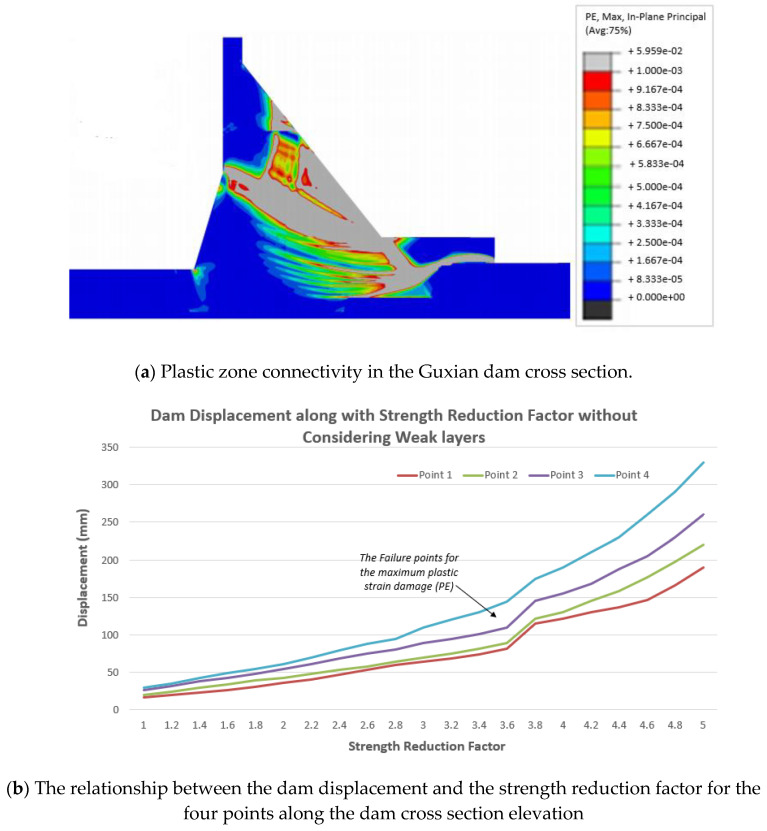
Guxian dam in the case of not considering the weak layers.

**Figure 12 materials-18-02893-f012:**
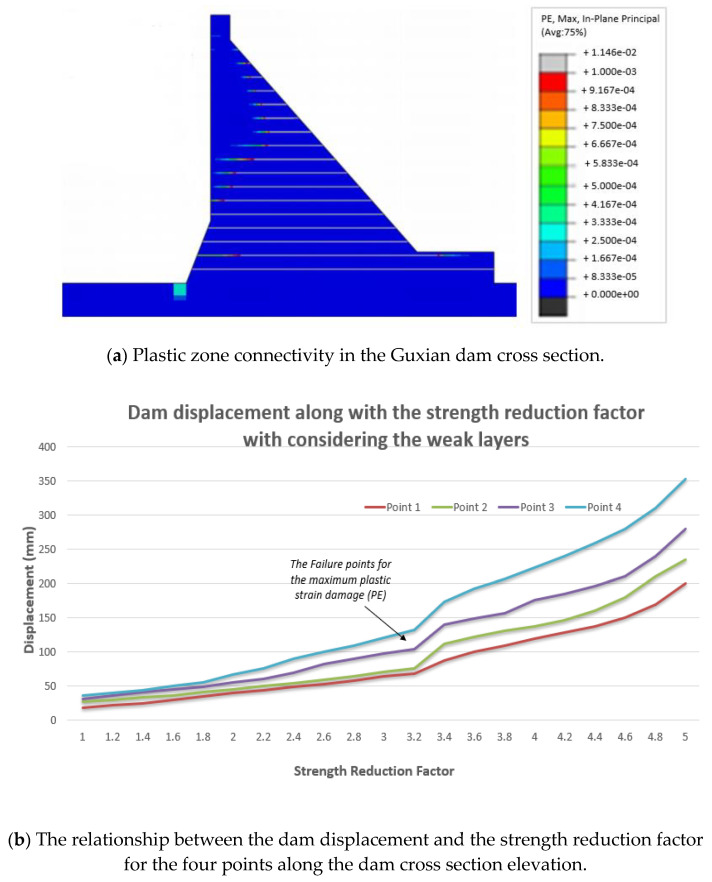
Guxian dam in the case of considering the weak layers.

**Figure 13 materials-18-02893-f013:**
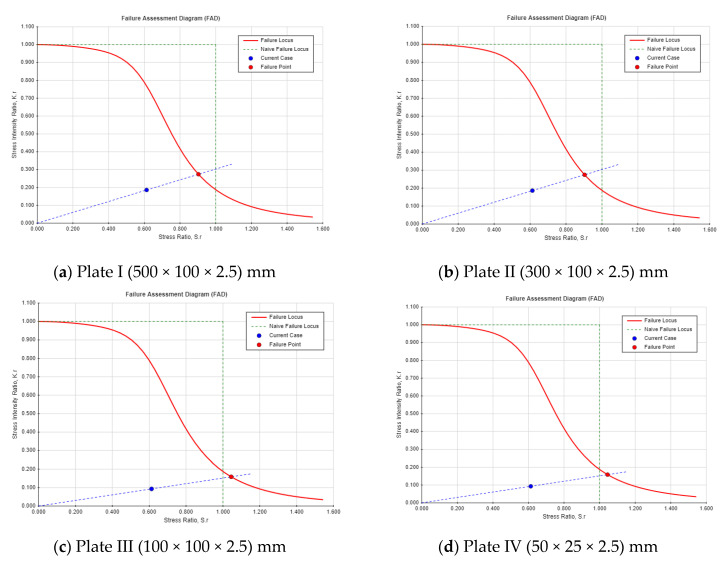
Failure assessment diagram (FAD) for the four plates in the case of center-through crack.

**Figure 14 materials-18-02893-f014:**
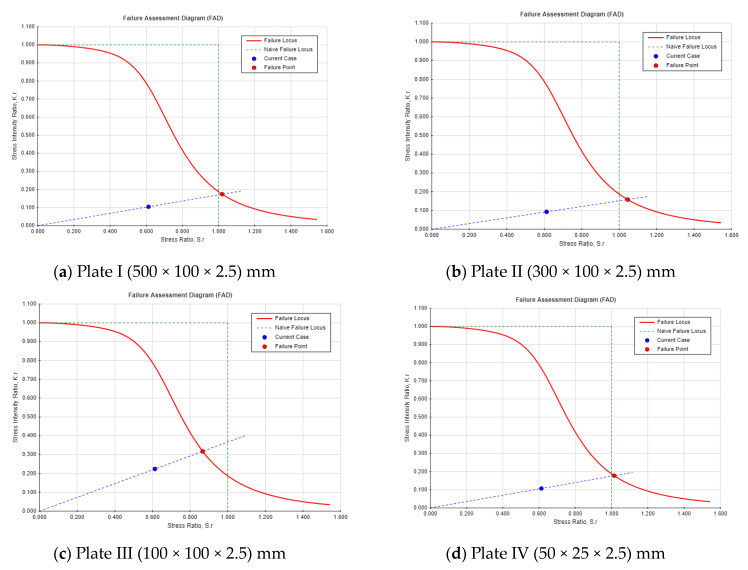
Failure assessment diagram (FAD) for the four plates in the single-edge crack.

**Figure 15 materials-18-02893-f015:**
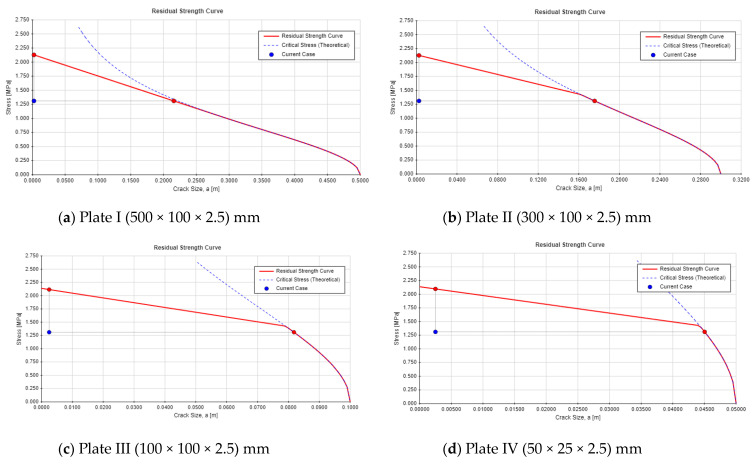
Residual strength curve for the center-through crack.

**Figure 16 materials-18-02893-f016:**
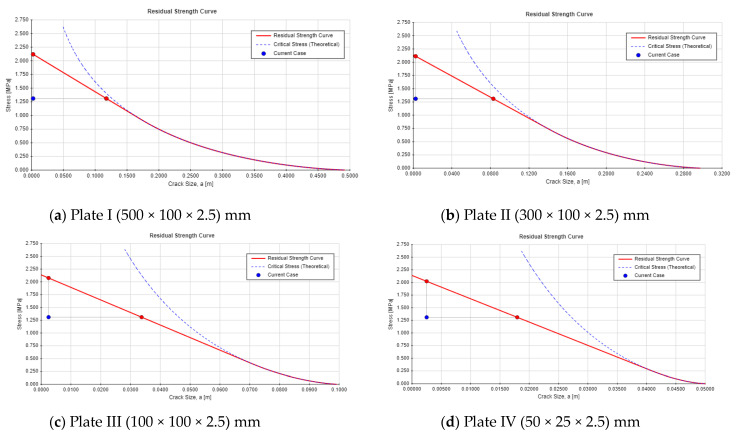
Residual strength curve for the single-edge crack.

**Figure 17 materials-18-02893-f017:**
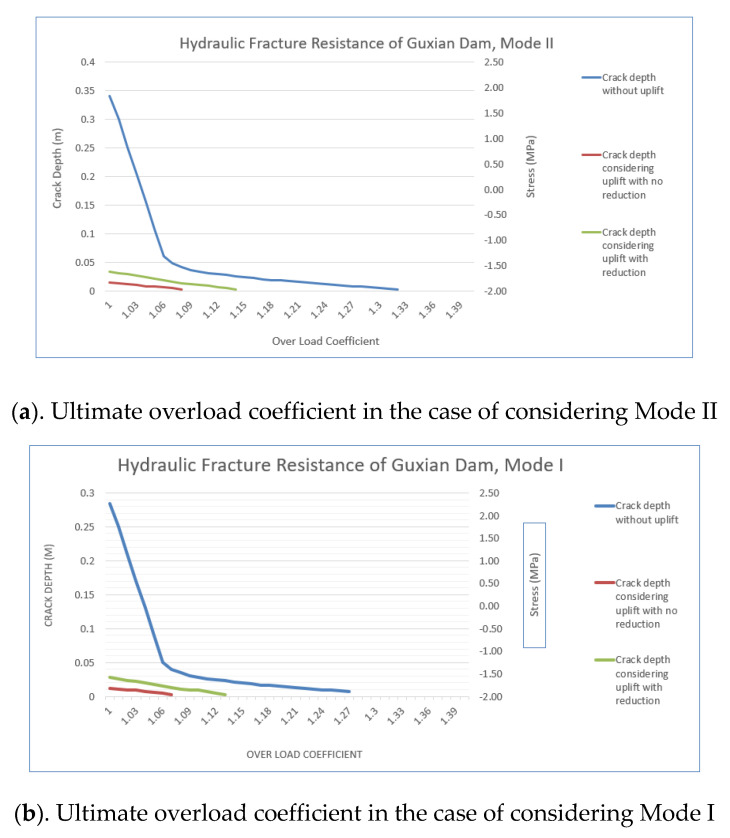
Guxian dam resistance to hydraulic fracture for the two modes of failure.

**Table 1 materials-18-02893-t001:** Stress intensity factors.

**Geometry**	**Stress Intensity Factor**
Center crack in an infinite body ** 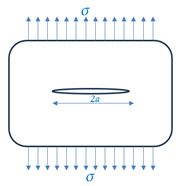 **	$K_{I}=\sigma \sqrt{\pi a}$
Single edge-through crack in semi-infinite body ** 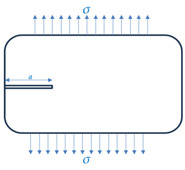 **	$K_{I}=1.12\sigma \sqrt{\pi a}$

**Table 2 materials-18-02893-t002:** The mechanical properties of the main materials used for the Guxian dam.

Material	Elastic Properties		Plastic Properties	
	Elastic Modulus GPa	Poisson Ratio	Friction Angle °	Yield Stress MPa
C15W6	22	0.167	50.7	1.78
BT CONCRETE	39	0.167	59.6	4.49
C20W6	25.5	0.167	51.3	2.48
C25W8	28	0.167	51.7	2.52
Foundation	32	0.2	50.7	1. 8

**Table 3 materials-18-02893-t003:** Material properties of the chosen plate according to the Guxian dam.

Mechanical Properties			Fracture Properties
Yield Strength (MPa)	Elastic Modulus (GPa)	Tensile Strength (MPa)	Fracture Toughness K_1c_ (MN/m^3/2^)
2.1	29.3	3	1.25

**Table 4 materials-18-02893-t004:** Dimensions of the plates under investigation.

Plate	b (mm)	t (mm)	a (mm)
1 (I)	500	100	2.5
2 (II)	300	100	2.5
3 (III)	100	100	2.5
4 (IV)	50	25	2.5

**Table 5 materials-18-02893-t005:** Stress field ahead a crack tip in a polar coordinate system.

	Mode I	Mode II
σ_xx_	$=\frac{K_{I}}{\sqrt{2\pi r}}\;\cos\left(\frac{\alpha }{2}\right)\left(1-\sin\left(\frac{\alpha }{2}\right)\sin\left(\frac{3\alpha }{2}\right)\right)$	$=\frac{K_{II}}{\sqrt{2\pi r}}\;\sin\left(\frac{\alpha }{2}\right)\left(2-\cos\left(\frac{\alpha }{2}\right)\cos\left(\frac{3\alpha }{2}\right)\right)$
σ_yy_	$=\frac{K_{I}}{\sqrt{2\pi r}}\;\cos\left(\frac{\alpha }{2}\right)\left(1+\sin\left(\frac{\alpha }{2}\right)\sin\left(\frac{3\alpha }{2}\right)\right)$	$=\frac{K_{II}}{\sqrt{2\pi r}}\;\sin\left(\frac{\alpha }{2}\right)\left(\cos\left(\frac{\alpha }{2}\right)\cos\left(\frac{3\alpha }{2}\right)\right)$
σ_xy_	$=\frac{K_{I}}{\sqrt{2\pi r}}\;\cos\left(\frac{\alpha }{2}\right)\left(\sin\left(\frac{\alpha }{2}\right)\cos\left(\frac{3\alpha }{2}\right)\right)$	$=\frac{K_{II}}{\sqrt{2\pi r}}\;\cos\left(\frac{\alpha }{2}\right)\left(1-\sin\left(\frac{\alpha }{2}\right)\sin\left(\frac{3\alpha }{2}\right)\right)$

**Table 6 materials-18-02893-t006:** List of parameters of the calculated section of the Guxian dam.

Parameter	Guxian	Dam Geometric Dimensions
*D*_1_ (m)	15	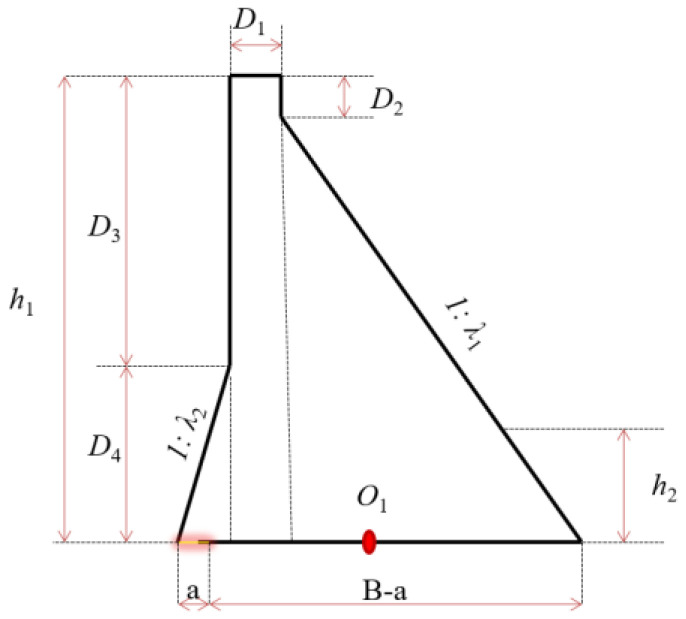
*D*_2_ (m)	20	
*D*_3_ (m)	165	
*D*_4_ (m)	50	
*λ* _1 (slope ratio)_	0.8	
*λ* _2 (slope ratio)_	0.3	
*h*_1_ (m)	215	
*h*_2_ (m)	25	
*B* (m)	172.35	
*γ*_wat_ (N/m^3^)	10,000	
*γ*_con_ (N/m^3^)	24,000	
*f*_t_ (MPa)	1.38	
*α*	0.5	

Note: the symbols indications are listed on the dam cross section in the same table.

**Table 7 materials-18-02893-t007:** Safety Indexes of Dam Displacement (mm).

Observation Point	Safety Index (In Terms of Displacement) Without Considering the Weak Layers (mm)	Safety Index (In Terms of Displacement) with Considering the Weak Layers (mm)
Point 1 (465 m)	115	88
Point 2 (480 m)	122	112
Point 3 (525 m)	145	140
Point 4 (631 m)	175	173

**Table 8 materials-18-02893-t008:** Safety monitoring indexes in case of crack propagation.

Plate/Crack Location	Center-Through Crack		Single-Edge Crack	
	FAD Safety Factor	Critical Crack Length (a_cr_) (m)	FAD Safety Factor	Critical Crack Length (a_cr_) (m)
1 (I)	1.71	0.2157	1.66	0.1176
2 (II)	1.71	0.17	1.66	0.08302
3 (III)	1.7	0.08	1.66	0.03372
4 (IV)	1.71	0.04	1.66	0.01795

## Data Availability

The original contributions presented in this study are included in the article. Further inquiries can be directed to the corresponding author.

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
