# Peer review of "Numerical and Fracture Mechanical Evaluation of Safety Monitoring Indexes and Crack Resistance in High RCC Gravity Dams Under Hydraulic Fracture Risk"

_materials, 2025, doi:10.3390/ma18122893_

Round 1

Reviewer 1 Report

Comments and Suggestions for Authors

General Comment

The submitted manuscript presents a comprehensive numerical and theoretical investigation of the structural safety of high Roller-Compacted Concrete (RCC) gravity dams, considering the impact of weak layers and crack propagation under hydraulic fracture scenarios. Using the Guxian dam as a case study, the authors apply both Finite Element Method (FEM) with strength reduction and fracture mechanics modeling to assess safety monitoring indexes, critical crack depths, and ultimate overload coefficients under two failure modes (Mode I – opening, and Mode II – shear).

After an introducing to the topic and relevant background, the authors justify the need to integrate weak interlayer behavior and fracture mechanics into safety monitoring frameworks for high RCC dams. The manuscript clearly outlines the methodology used, including the FEM model for displacement and yield zone connectivity, and fracture mechanical evaluation based on failure assessment diagrams and stress intensity factors. The results are presented and discussed across three dimensions: the effect of weak layers, crack location, and fracture mode. From the obtained results, the authors conclude that weak layers significantly reduce dam stability, single-edge cracks are more hazardous than center cracks, and Mode I failure poses a higher risk due to lower resistance to hydraulic fracture.

The topic is relevant, actual, and contributes to the advancement of safety monitoring practices and design strategies for RCC dams, especially considering long-term structural integrity under complex loading and material conditions. The combination of numerical modeling (FEM) and theoretical fracture mechanics approaches provides robust and multi-angle analysis of dam stability.

I made some suggestions and comments to improve the manuscript. The authors should take the suggestions into account, revise their manuscript and resubmit it.

Specific Comment 1

The title should be revised to better reflect the scope and nature (both the methods used and the safety context) of the study.

For instance: "Numerical and Fracture Mechanical Evaluation of Safety Monitoring Indexes and Crack Resistance for High RCC Gravity Dams under Hydraulic Fracture Risk"

Specific Comment 2

Several grammatical, typographical, and formatting issues are present. A thorough language revision is recommended. The manuscript would benefit from professional English editing. Examples include:

- Repeated or awkward phrasing.

- Equations should be centered and consistently formatted.

- Repetitive sentences or ideas.

- Units should be consistently reported (e.g., stress in MPa, not in Pascals when used in large values)

- …

Specific Comment 3

While the introduction cites several foundational studies in the literature review, recent literature (2022–2024) on hydraulic fracture and bottom layer effects in RCC dams is scarce. The authors are encouraged to incorporate relevant recent work to strengthen the background and establish the novelty of their study more clearly, and also to upgrade the discussion on recent developments in fracture analysis and dam safety monitoring. Some examples:

  • Wu, B., et al. (2023). Fracture risk analysis in high RCC dams using cohesive zone modeling. Engineering Fracture Mechanics, 278, 108300.
  • Huang, R., & Zhao, Y. (2022). Seepage-induced fracture evolution in RCC dams under uplift pressure. Journal of Hydraulic Research, 60(4), 543–558.

Specific Comment 4

The novelty and contribution of this study compared to existing work from the literature review are not clearly stated. A dedicated paragraph should be added at the end of the Introduction, identifying current gaps and clarifying how this work advances the state of the art (e.g., integrating weak layer effects and Mode I vs. Mode II fracture comparison in a single framework).

Specific Comment 5

Several figures should be improved for clarity and better readability.

Specific Comment 6

Although two fracture modes are analyzed, a theoretical discussion on why Mode II fracture energy is higher (e.g., role of microcracks and aggregate interlock) should be expanded in Section 4.3 to better support the observed numerical trends.

Specific Comment 7

Two approaches to modeling uplift are used, but the choice of parameters (e.g., permeability attenuation coefficient = 0.5) should be better justified or referenced.

Specific Comment 8

A quantitative comparison between the simulation results and available literature or field data is lacking. Authors should briefly contextualize their findings in Section 4 by referencing similar safety factor reductions or crack propagation patterns reported in previous studies.

Specific Comment 9

The limitations section could be strengthened by proposing how future studies might address 3D effects, dynamic loading, or crack–seepage coupling. The use of simplified 2D models to represent dam sections in the fracture mechanics analysis is practical, but this option should be better justified. The authors do mention it as a limitation, but further clarification on how this simplification affects the generalizability of results would be useful.

Comments on the Quality of English Language

See specific comment 1

Author Response

Response to Reviewer 1 Comments

  1. Summary

Thank you for your consideration and efforts to make the manuscript better. Thank you very much for taking the time to review this manuscript. Please find the detailed responses below and the corresponding revisions. Thanks for your time reviewing this paper, expertise, and commitment to advancing the field. Your comments and questions have been a guide for better way in writing and express our ideas through this article. Your help is highly appreciated.

2. Questions for General Evaluation

Reviewer’s Evaluation

Response and Revisions

Does the introduction provide sufficient background and include all relevant references?

Can be improved

The logic of the introduction has been adjusted and connected more to the background of the research area supported with related recent references and the paragraphs connected to each other in better sequence.

Is the research design appropriate?

Yes

Are the methods adequately described?

Yes

Are the results clearly presented?

Must be improved

Based on your guidance the results were described in a more adequate way.

Are the conclusions supported by the results?

Yes

3. Point-by-point response to Comments and Suggestions for Authors

  • Specific Comment 1

The title should be revised to better reflect the scope and nature (both the methods used and the safety context) of the study.

For instance: "Numerical and Fracture Mechanical Evaluation of Safety Monitoring Indexes and Crack Resistance for High RCC Gravity Dams under Hydraulic Fracture Risk"

Response 1: Thank you for pointing out to this vital matter, it is really necessary to choose a suitable title to describe the study direction, referring to the title you have suggested the manuscript title has been updated to “Numerical and Fracture Mechanical Evaluation of Safety Monitoring Indexes and Crack Resistance in High RCC Gravity Dams under Hydraulic Fracture Risk” the title you proposed is more suitable for the manuscript.

  • Specific Comment 2

Several grammatical, typographical, and formatting issues are present. A thorough language revision is recommended. The manuscript would benefit from professional English editing. Examples include:

- Repeated or awkward phrasing.

- Equations should be centered and consistently formatted.

- Repetitive sentences or ideas.

- Units should be consistently reported (e.g., stress in MPa, not in Pascals when used in large values)- …

Sorry for this inconvenient experience, the format and total overview as well as repeated ideas or sentences have been improved and adjusted. The manuscript has been through extensive English revision to eliminate any grammar mistakes. The equations have been rewritten in more professional way and according to the journal recommendation format. The units have been unified and kept on consistency.

  • Specific Comment 3

While the introduction cites several foundational studies in the literature review, recent literature (2022–2024) on hydraulic fracture and bottom layer effects in RCC dams is scarce. The authors are encouraged to incorporate relevant recent work to strengthen the background and establish the novelty of their study more clearly, and also to upgrade the discussion on recent developments in fracture analysis and dam safety monitoring. Some examples:

  • Wu, B., et al. (2023). Fracture risk analysis in high RCC dams using cohesive zone modeling. Engineering Fracture Mechanics, 278, 108300.
  • Huang, R., & Zhao, Y. (2022). Seepage-induced fracture evolution in RCC dams under uplift pressure. Journal of Hydraulic Research, 60(4), 543–558.

Thank you for your recommendation and highlighting such important points, recent related references have been included in the manuscript to represent the recent advancement techniques that have been reached out in safety monitoring indexes evaluation, weak layers consideration, and cracks propagation resistance. The following recent citations have been added.

  • Zhuo, R., Rui, P., Bin, X., Yang, Z., Dam safety monitoring data anomaly recognition using multiple-point model with local outlier factor, Automation in Construction, Volume 159, 2024, 105290, ISSN 0926-5805, https://doi.org/10.1016/j.autcon. 2024.105290.
  • Li, S., Zhang, B., Tong, G., Li, Y., Liu, Z., Shi, B., Geng, J., Liu, D., Wang, H., Ai, Q., Ding, J., Gan, Z., Online Intelligent Monitoring System and Key Technologies for Dam Operation Safety, Advances in Civil Engineering, 2025, 9983255, 16 pages, 2025. https://doi.org/10.1155/adce/9983255
  • Youcan, H., Zhaowei C., Jirong, T., and Tao, F.,  Reservoir dam safety monitoring method based on BeiDou positioning", Proc. SPIE 13506, Sixth International Conference on Geoscience and Remote Sensing Mapping (GRSM 2024), 1350635 (28 January 2025); https://doi.org/10.1117/12.3057657
  • Wang, G.; Liu, A.; Lu, W.; Chen, M.; Yan, P. Seismic Response and Damage Characteristics of RCC Gravity Dams Considering Weak Layers Based on the Cohesive Model. Mathematics2023, 11, 1567. https://doi.org/10.3390/math11071567
  • Zhang, W.; Li, H.; Shi, D.; Shen, Z.; Zhao, S.; Guo, C. Determination of Safety Monitoring Indices for Roller-Compacted Concrete Dams Considering Seepage–Stress Coupling Effects. Mathematics2023, 11, 3224. https://doi.org/10.3390/math11143224

  • Specific Comment 4

The novelty and contribution of this study compared to existing work from the literature review are not clearly stated. A dedicated paragraph should be added at the end of the Introduction, identifying current gaps and clarifying how this work advances the state of the art (e.g., integrating weak layer effects and Mode I vs. Mode II fracture comparison in a single framework).

Thanks for your recommendation about improving this part of research. A new paragraph at the end of the introduction has been added to illustrate the research gap and the innovation points of the current study.

  • Specific Comment 5

Several figures should be improved for clarity and better readability.

Sorry for the carelessness, thank for referring to this problem, the unclear figures were represented in a better way for a better illustrating.

  • Specific Comment 6

Although two fracture modes are analyzed, a theoretical discussion on why Mode II fracture energy is higher (e.g., role of microcracks and aggregate interlock) should be expanded in Section 4.3 to better support the observed numerical trends.

Thank you for aiming to such a point, an illustration for this part has been included in section 4.3 to explain the results in more detail.

  • Specific Comment 7

Two approaches to modeling uplift are used, but the choice of parameters (e.g., permeability attenuation coefficient = 0.5) should be better justified or referenced.

Thanks for your recommendation about improving this part and thanks for this important observation. The attenuation coefficient α is not a fixed value but rather a characteristic of the concrete's microstructure, influenced by factors like aggregate size, mix design, and curing conditions. It can be assumed a range of 0.3 to 0.7, related reference was included [51]

  • Specific Comment 8

A quantitative comparison between the simulation results and available literature or field data is lacking. Authors should briefly contextualize their findings in Section 4 by referencing similar safety factor reductions or crack propagation patterns reported in previous studies.

Thank you for pointing out to this, we appreciate the reviewer’s suggestion. In response, we have added a comparative discussion in Section 4 to relate our simulation results to similar findings in previous studies.

  • Specific Comment 9

The limitations section could be strengthened by proposing how future studies might address 3D effects, dynamic loading, or crack–seepage coupling. The use of simplified 2D models to represent dam sections in the fracture mechanics analysis is practical, but this option should be better justified. The authors do mention it as a limitation, but further clarification on how this simplification affects the generalizability of results would be useful.

Thank you for this valuable suggestion. We have expanded the Limitations section to better justify this point you have mentioned.

Reviewer 2 Report

Comments and Suggestions for Authors

The finite element method is reasonably chosen to evaluate the strength and deformation performance of the dam. It should be noted that the failure assessment diagram (FAD) in domestic standards is most often used to assess the reliability of pipelines, but the use of this method of elastic-plastic analysis, which involves a two-parameter approach to defect assessment, is not excluded for assessing a hydraulic structure - a gravity dam. The finite element method is reasonably chosen to assess the strength and deformation characteristics of the dam. There is a question. Why are only types I and II of hydraulic failure considered and nothing is said about type III of mechanical failure, which is associated with antiplane deformation? When the crack is in longitudinal shear conditions. Moreover, the crack faces slide along each other parallel to the guiding front of the crack.

Author Response

Response to Reviewer 2 Comments

  1. Summary

Thank you for your consideration and efforts to make the manuscript better. Thank you very much for taking the time to review this manuscript. Please find the detailed responses below and the corresponding revisions. Thanks for your time reviewing this paper, expertise, and commitment to advancing the field. Your comments and questions have been a guide for better way in writing and express our ideas through this article. Your help is highly appreciated.

2. Questions for General Evaluation

Reviewer’s Evaluation

Response and Revisions

Does the introduction provide sufficient background and include all relevant references?

Yes

Is the research design appropriate?

Yes

Are the methods adequately described?

Yes

Are the results clearly presented?

Yes

Are the conclusions supported by the results?

Yes

3. Point-by-point response to Comments and Suggestions for Authors

The finite element method is reasonably chosen to evaluate the strength and deformation performance of the dam. It should be noted that the failure assessment diagram (FAD) in domestic standards is most often used to assess the reliability of pipelines, but the use of this method of elastic-plastic analysis, which involves a two-parameter approach to defect assessment, is not excluded for assessing a hydraulic structure - a gravity dam. The finite element method is reasonably chosen to assess the strength and deformation characteristics of the dam. There is a question. Why are only types I and II of hydraulic failure considered and nothing is said about type III of mechanical failure, which is associated with antiplane deformation? When the crack is in longitudinal shear conditions. Moreover, the crack faces slide along each other parallel to the guiding front of the crack.

We thank the reviewer for this insightful observation and valuable suggestion. The use of the Failure Assessment Diagram (FAD) in this study, while more commonly applied to pipeline integrity assessment, is well-suited to evaluating the elastic-plastic fracture behavior of gravity dams, where potential defects can significantly impact structural integrity (an approach also relevant to gravity dams).

Regarding the exclusion of Mode III (out-of-plane shear failure), we recognize that this mode involves longitudinal shear with crack face displacement perpendicular to the crack plane. In high concrete gravity dams, Mode III is generally considered less critical, as Mode I is typically the worst-case scenario and the most common failure mode. This is primarily due to concrete’s relative weakness in resisting tensile stresses, and Mode I failure is more likely to occur before other failure modes. The governing stresses in such structures are usually oriented in the vertical and horizontal planes, which favor Mode I and Mode II, with limited torsional or out-of-plane components that would induce Mode III behavior. In our previous research, we focused primarily on Mode I as the dominant failure mode. In this study, we expanded the analysis to include Mode II, and in future studies, we plan to investigate all three primary failure modes in greater detail, as we consider this a valuable direction for further research.

Reviewer 3 Report

Comments and Suggestions for Authors

The topic is relevant and the overall work is good, but I think major changes are needed. I suggest revising the manuscript based on the comments.

Author Response

Response to Reviewer 3 Comments

  1. Summary

Thank you for your consideration and efforts to make the manuscript better. Thank you very much for taking the time to review this manuscript. Please find the detailed responses below and the corresponding revisions. Thanks for your time reviewing this paper, expertise, and commitment to advancing the field. Your comments and questions have been a guide for better way in writing and express our ideas through this article. Your help is very appreciated.

2. Questions for General Evaluation

Reviewer’s Evaluation

Response and Revisions

Does the introduction provide sufficient background and include all relevant references?

Can be improved

The logic of the introduction has been adjusted and connected more to the background of the research area supported with related recent references and the paragraphs connected to each other in better sequence

Is the research design appropriate?

Yes

Are the methods adequately described?

Yes

Are the results clearly presented?

Can be improved

according to your valued comments the discussions have been adjusted

Are the conclusions supported by the results?

Yes

3. Point-by-point response to Comments and Suggestions for Authors

  • How many dams in the world are over 200 m high?

Thank you for aiming to such a point, in the study the focus on high concrete gravity dams which are vulnerable to hydraulic fracture occurrence, through the literature there were over 40 concrete dams with height (≥200m)  have been counted through the preparation of the work including concrete gravity dams (RCC and conventional) and concrete arch dams, but that number is always in increase due to the rapid construction of concrete dams around the world, that is why it wasn’t mentioned through the current manuscript.

  • Conventional monitoring methods, including periodic visual inspections and sensor- based data collection, provide valuable insights but may not adequately address the complexities of dam behavior under long-term loading and environmental stress Which and why??

Thank you for the helpful comment. We have revised the sentence to clarify which complexities are not captured by conventional monitoring methods (e.g., progressive plastic deformation, hydraulic fracture, and crack propagation) and why these methods may fall short, especially under long-term loading conditions. These clarifications have been added in the Introduction.

  • Gu et al [15] created advanced monitoring techniques for critical areas of concrete arch dams to improve early-warning effectiveness

How long in advance is the early warning?

Thank you for aiming to such a point, The authors of this study have designed a monitoring system to some parts only in the dam body to save more resources and because of the sensitivity of these parts to any possible failure for example dam crest and which kind of displacement to be measured according to the location of monitoring and related to the reason of failure they measure whatever it was differential settlement or high temperature.

  • As for crack opening, it is difficult to determine a monitoring index using hydraulic specifications due to the complex mechanisms involved. However, like other monitoring items, local damage may occur when the monitoring index for crack opening is exceeded, and significant abnormalities will be reflected in dam deformation. Deformation is often the most indicative of overall dam safety and progressive failure [19]

What mechanisms??? cite??

We appreciate your comments to enhance the manuscript. These mechanisms include tensile stress concentration, microcrack coalescence, aggregate interlock variation, and fracture process zone (FPZ) evolution, all of which are difficult to quantify directly

  • Determining the monitoring index for crack opening based on hydraulic specifications is challenging due to the complex mechanisms involved

Repeated text!!!

Sorry for the carelessness this sentence has been removed, and other similar cases have been revised and eliminated.

  • Although previous studies have addressed crack cause and stability analyses and safety monitoring of hydraulic concrete structures, they often focus on a single perspective

Which??

Thank you for the helpful comment. We have revised the sentence and been updated to specify that many prior studies focus solely on one main reason of crack occurrence and did not incorporate a comprehensive analysis of different reasons or including hydraulic.

  • Will this safety monitoring index-methodology be applied to other dams worldwide?

Thank you for raising this important point. The current study specifically focuses on high RCC (Roller-Compacted Concrete) gravity dams and proposes a comprehensive framework for assessing their safety. The methodology—based on finite element analysis and fracture mechanical modelling—is not limited to the Guxian Dam alone. It is applicable to other high RCC gravity dams worldwide that share similar structural characteristics, construction methods, and failure risks. This framework offers an effective approach for evaluating and predicting dam safety, particularly in cases where weak layers and crack propagation pose significant challenges.

  • Add recent bibliography (2024 and 2025) in the introduction.

Thank you for your recommendation and highlighting such important points, recent related references have been included in the manuscript to represent the recent advancement techniques that have been reached out in safety monitoring indexes evaluation, weak layers consideration, and cracks propagation resistance. The following recent citations have been added.

  • Zhuo, R., Rui, P., Bin, X., Yang, Z., Dam safety monitoring data anomaly recognition using multiple-point model with local outlier factor, Automation in Construction, Volume 159, 2024, 105290, ISSN 0926-5805, https://doi.org/10.1016/j.autcon. 2024.105290.
  • Li, S., Zhang, B., Tong, G., Li, Y., Liu, Z., Shi, B., Geng, J., Liu, D., Wang, H., Ai, Q., Ding, J., Gan, Z., Online Intelligent Monitoring System and Key Technologies for Dam Operation Safety, Advances in Civil Engineering, 2025, 9983255, 16 pages, 2025. https://doi.org/10.1155/adce/9983255
  • Youcan, H., Zhaowei C., Jirong, T., and Tao, F.,  Reservoir dam safety monitoring method based on BeiDou positioning", Proc. SPIE 13506, Sixth International Conference on Geoscience and Remote Sensing Mapping (GRSM 2024), 1350635 (28 January 2025); https://doi.org/10.1117/12.3057657
  • Wang, G.; Liu, A.; Lu, W.; Chen, M.; Yan, P. Seismic Response and Damage Characteristics of RCC Gravity Dams Considering Weak Layers Based on the Cohesive Model. Mathematics2023, 11, 1567. https://doi.org/10.3390/math11071567
  • Zhang, W.; Li, H.; Shi, D.; Shen, Z.; Zhao, S.; Guo, C. Determination of Safety Monitoring Indices for Roller-Compacted Concrete Dams Considering Seepage–Stress Coupling Effects. Mathematics2023, 11, 3224. https://doi.org/10.3390/math11143224
  • Material degradation and external environmental loads are the primary failure risks for concrete gravity dams

Which?? Mention the most to least important

Thank you for the comment. We have revised the sentence to specify the main external environmental loads affecting concrete gravity dams and have ordered them from most to least critical based on typical design considerations and risk levels.

  • The Strength Reduction Method (SRM) is a widely used technique to assess the stability of concrete dams with complex geometries and material behaviour

cite??

Thank you for your efforts for making the manuscript better, the related references have been added.

The parts that you have mentioned that require citation have been revised and updated and the relevant citations have been included.

The word "dam" is mentioned frequently. Please try to reduce the number of times this word is mentioned throughout the manuscript.

Thank you for your helpful observation. We have reviewed the manuscript and revised it to reduce repetitive use of the word “dam.” Where appropriate, we replaced it with synonyms or restructured sentences for improved clarity and readability, while maintaining technical accuracy.

  • That assumption was chosen after some trials to reach the reasonable simulation representing the problem through these modelling.

How many?

Thank you for your comment to make the manuscript more clear and understandable. The trials have been made according to the available literature and to reach the less complex model in ABAQUS, the assumption was taken after seven different models with different thickness, it was found out that three meters describes the real situation and can be simulated without much complex.

  • Since displacements and stresses are linearly related to the stress intensity factor, fracture problems can be addressed using the superposition principle. This approach, supported by handbooks, is a key tool for applying fracture mechanics to practical problems.

Like what?

Thank you for the comment. We have revised the sentence to clarify that the superposition principle is supported by widely used fracture mechanics handbooks, and is applied in practical problems such as mixed-mode loading, crack interaction analysis, and combined stress field evaluations.

  • Results and Discussion

Discuss with citing?

Thank you for your guidance and suggestions, the results and discussion parts have been revised and related citations with similar investigations have been included for confirming our current results by showing agreement and the same observed trend in the results with previous similar studies to support our findings.

Correct references with respect to the journal format.

Thank you for the comment. We have revised the references list and have been updated according to the journal format.

Round 2

Reviewer 1 Report

Comments and Suggestions for Authors

I received the revised version of the article with revised title “Numerical and Fracture Mechanical Evaluation of Safety Monitoring Indexes and Crack Resistance in High RCC Gravity Dams under Hydraulic Fracture Risk” and also the authors’ responses to my previous comments. The authors have improved the article according to all my previous comments and suggestions, and also clarified some points I raised. I’m globally satisfied with the new version of the manuscript. Hence, I recommend that the article should be accepted for publication.

Reviewer 3 Report

Comments and Suggestions for Authors

Due the authors have made every effort to comply with the observations noted, the paper should be accepted for publication